# Time-Aware Prior Fitted Networks for Zero-Shot Forecasting with Exogenous Variables

**Andres Potapczynski**[*]                                      *ap6604@nyu.edu*
*Amazon Science, New York University*

**Ravi Kiran Selvam**[*]                                        *ravisel@amazon.com*
*Amazon Science*

**Tatiana Konstantinova**                                      *tkonst@amazon.com*
*Amazon Science*

**Malcolm Wolff**                                              *wolfmalc@amazon.com*
*Amazon Science*

**Kin G. Olivares**                                            *kigutie@amazon.com*
*Amazon Science*

**Ruijun Ma**                                                  *ruijunma@amazon.com*
*Amazon Science*

**Michael W. Mahoney**                                         *zmahmich@amazon.com*
*Amazon Science*

**Andrew G. Wilson**                                           *wilsmman@amazon.com*
*Amazon Science, New York University*

**Boris N. Oreshkin**                                          *oreshkin@amazon.com*
*Amazon Science*

**Dmitry Efimov**                                              *defimov@amazon.com*
*Amazon Science*

**Reviewed on OpenReview:** *https://openreview.net/forum?id=nJARpxp3cF*

## Abstract

In many time series forecasting settings, the target time series is accompanied by exogenous covariates, such as promotions and prices in retail demand; temperature in energy load; calendar and holiday indicators for traffic or sales; and grid load or fuel costs in electricity pricing. Ignoring these exogenous signals can substantially degrade forecasting accuracy, particularly when they drive spikes, discontinuities, or regime and phase changes in the target series. Most current time series foundation models (e.g., `Chronos`, `Sundial`, `TimesFM`, `TimeMoE`, `TimeLLM`, and `LagLlama`) ignore exogenous covariates and make forecasts solely from the numerical time series history, thereby limiting their performance. In this paper, we develop `ApolloPFN`, a prior-data fitted network (PFN) that is time-aware (unlike prior PFNs) and that natively incorporates exogenous covariates (unlike prior univariate forecasters). Our design introduces two major advances: (i) a synthetic data generation framework that injects realistic temporal patterns, structural changes, and exogenous dependencies into the PFN prior; and (ii) time-aware architectural modifications that embed inductive biases needed to exploit temporal context. We demonstrate that `ApolloPFN` outperforms existing

---

[*]Equal contribution

baselines across several forecasting benchmarks with exogenous information, including M5, electric price forecasting, UCI Air Quality, and Solar Energy datasets.

# 1 Introduction

In many high-impact forecasting scenarios, leveraging *exogenous information*, i.e., inputs beyond the raw numerical target time series values, is essential. For example, in electricity price forecasting and consumer demand forecasting, information about planned prices and promotions, merchandising changes, holidays and local events, weather forecasts, and competitor pricing, are naturally encoded categorically and can shift demand sharply. Ignoring this information often induces large, systematic errors. This is illustrated in Figure 1. Despite the clear value of exogenous information, most existing time series foundation models (TSFMs), including `Chronos` (Ansari et al., 2024), `Sundial` (Liu et al., 2025), `TimesFM` (Das et al., 2023), `TimeMoE` (Shi et al., 2025), `TimeLLM` (Jin et al., 2024), and `LagLlama` (Rasul et al., 2023) either cannot incorporate exogenous covariates directly or require task-specific fine-tuning (Arango et al., 2025; Wang et al., 2024; Potapczynski et al., 2024). Such fine-tuning is often undesirable: it adds runtime overhead, complicates inference pipelines, increases deployment costs, and weakens the anonymity and isolation of downstream customer data. Consequently, zero-shot capability is an important requirement for a TSFM serving as a substrate for downstream systems and products (Bommasani et al., 2021). Therefore, a practical modern TSFM should natively incorporate accompanying exogenous covariates whenever they are available.

There are a few foundation-like models that support exogenous covariates in a zero-shot time series forecasting setting, most notably `TabPFN-TS` (Hoo et al., 2025), `Moirai` (Woo et al., 2024) and more recently `Chronos-2` (Ansari et al., 2025). These methods represent two related but distinct directions for foundation models in time series forecasting. `Moirai` and `Chronos2` follow the paradigm of large-scale pretrained forecasting models trained on diverse time series corpora, whereas `TabPFN-TS` builds on the prior-data fitted network (PFN) framework, where predictions are generated by conditioning directly on the observed context of the test input. Because large pretrained models such as `Moirai` and `Chronos2` are trained on broad collections of public time series datasets, evaluating fully non-overlapping train/test exposure can be challenging in practice, especially for widely used benchmark suites such as Monash datasets (Makridakis & Hibon, 1979; Makridakis et al., 2020; 2022). Nevertheless, both paradigms have demonstrated strong zero-shot forecasting capability under different settings.

However, `TabPFN-TS` does not explicitly introduce temporal architectural inductive biases; rather, it extends the tabular PFN framework by incorporating a limited set of time series-specific features. Consequently, it inherits inductive biases optimized for i.i.d. tabular prediction settings, which may not fully align with the sequential dependencies and temporal dynamics inherent in forecasting problems (Yu et al., 2025b;a) In particular, its architecture exhibits substantial invariance to observation order, which can limit its ability to fully capture temporal structure and recency effects. These limitations can manifest in weaker handling of order-dependent dynamics, reduced robustness across unseen frequencies, less reliable trend extrapolation, and weaker uncertainty calibration in some forecasting settings. Motivated by these limitations, we develop a PFN framework with explicit temporal inductive biases that preserves the zero-shot inference and covariate-conditioning capabilities of PFNs while better capturing the structure of time series data.

In this paper, we study how to effectively leverage exogenous variables for zero-shot time series forecasting. Our main contributions are as follows:

- We provide a systematic analysis of the limitations of existing PFN-based approaches, including `TabPFN-TS`, when applied to time series forecasting. Specifically, we show that the i.i.d. assumptions underlying the synthetic task distribution and architectural design choices inherited from tabular PFNs limit their ability to explicitly capture temporal dependencies, such as autocorrelation, ordered patterns, and evolving dynamics. These limitations can lead to degraded performance on forecasting problems requiring strong temporal reasoning, as illustrated in Figure 1 (b).

- We introduce `ApolloPFN`, a time-aware PFN framework designed to address these limitations while retaining the zero-shot inference and covariate-conditioning capabilities of PFNs (Section 3). `ApolloPFN`

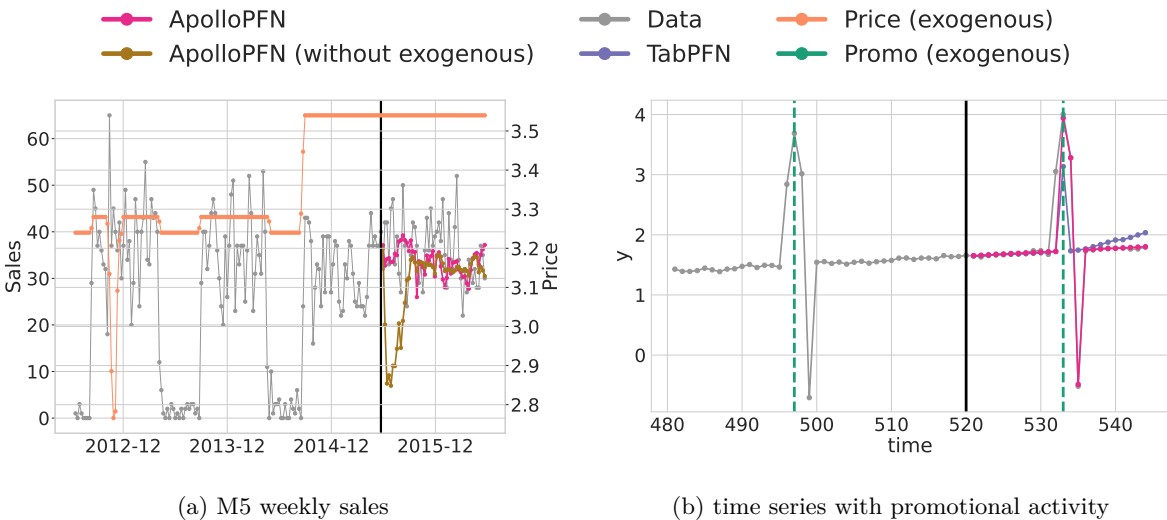

(a) M5 weekly sales

(b) time series with promotional activity

Figure 1: (a) **Not using exogenous information can lead to catastrophic forecasting errors.** We compare the predictions of `ApolloPFN` with and without exogenous information for the weekly sales of a real product from the M5 benchmark. Ignoring the price information leads the forecaster to predict a decreased demand via context parroting (brown), whereas the tracking of price dynamics helps the model focus on most up-to-date dynamics (pink). (b) **Prior-data fitted networks such as `TabPFN-TS` fail to capture ordered patterns.** We compare the prediction of `TabPFN-TS` and `ApolloPFN` for a synthetic time series that has a recurrent pattern of a ramp-up period before a promotion, a spike on the promotion, a ramp-down period, and then a subsequent decrease in demand. The exogenous promotion information is encoded as a binary indicator. Training data is to the left of the black line, and forecasts are to the right.

introduces two complementary components: (i) a synthetic time series data generation procedure based on a novel graph generation algorithm that improves prior construction efficiency and accelerates learning (Figure 4) and incorporates time-dependent root nodes (Section 3.2); and (ii) architectural inductive biases that explicitly encode temporal ordering and dependencies (Section 3.3).

- We conduct extensive evaluations of `ApolloPFN` against competitive baselines, including `TabPFN-TS` and `Moirai`, in several datasets spanning more than 90K time series *that have accompanying exogenous covariates*, as well as standard benchmarks without exogenous covariates, demonstrating the broad effectiveness of `ApolloPFN` (Section 4). We present several ablations on real and synthetic data benchmarks to bolster our architectural and data generation choices (Section 4.4).

## 2 Limitations of `TabPFN-TS` for Time Series Forecasting

`TabPFN-TS` (Hoo et al., 2025) extends the tabular foundation model `TabPFN-v2` (Hollmann et al., 2025) by incorporating a set of task-specific, manually engineered time series features to enable forecasting. While `TabPFN-TS` achieves competitive performance on several standard forecasting benchmarks, its design inherits inductive biases from tabular PFNs that are not explicitly tailored to sequential data. In this section, we analyze several representative limitations of this approach, as illustrated in Figure 1 and Figure 2.

**Inability to learn ordered patterns.** Ordered seasonal patterns that span across multiple time steps are very common in industry applications such as demand forecasting where a product has a gradual increase in demand until its promotion date and sharply drops after the promotion. These types of patterns are not purely cyclical, but instead they reflect structured temporal dependencies that unfold over multiple horizons. An example of such pattern is shown in Figure 1(b), which shows that `TabPFN-TS` cannot capture it via in-context, as it lacks appropriate temporal inductive biases. Instead, the model resolves to outputting a smaller spike in the promotional event.

**Dependency on manually engineered frequency features.** `TabPFN-TS` relies on running index feature as well as frequency features that are taken from the timestamp of the data (such as day-of-week, day-of-month, month-of-year, etc.) or estimated frequencies obtained through a FFT decomposition of the time

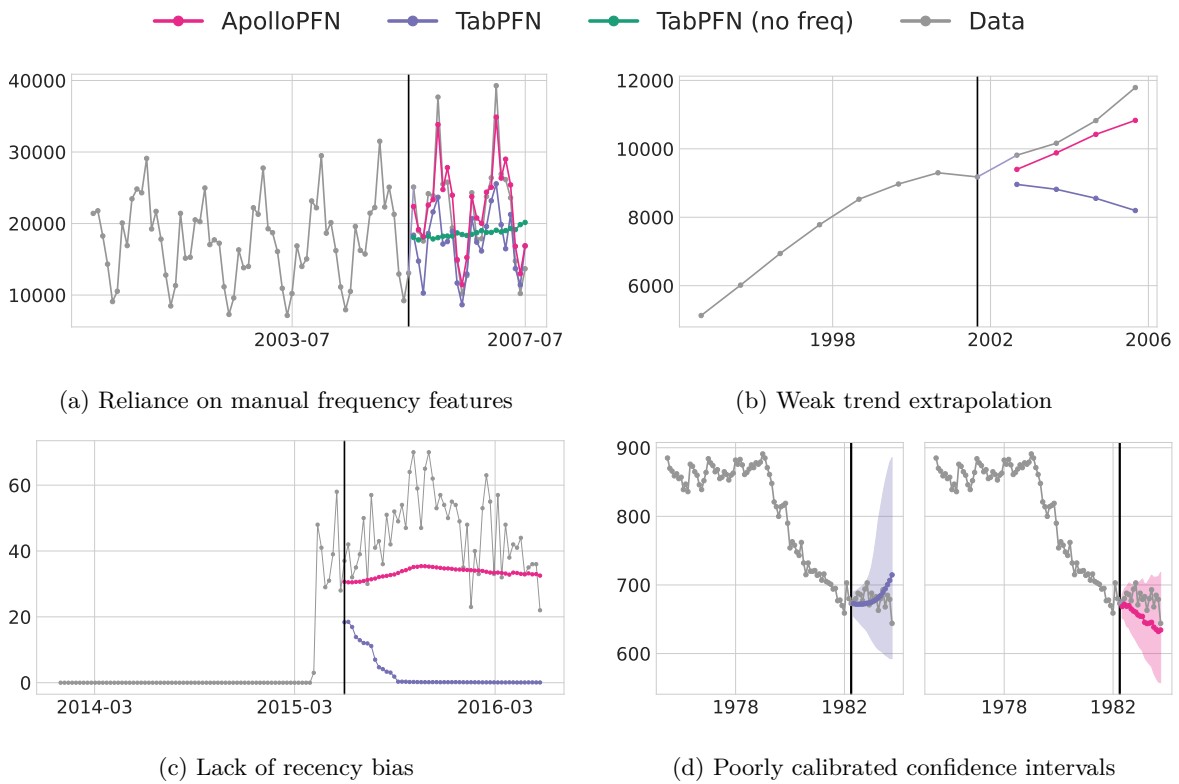

(a) Reliance on manual frequency features

(b) Weak trend extrapolation

(c) Lack of recency bias

(d) Poorly calibrated confidence intervals

Figure 2: **Limitations of `TabPFN-TS` for time series forecasting and corresponding improvements in `ApolloPFN`.** We provide illustrative examples of each of the identified limitations using real-world time series from different datasets: Tourism Monthly for (a), Tourism Yearly for (b), M5 Weekly for (c), and M1 Monthly for (d). In all plots, the in-context historical observations are shown to the left of the black vertical line, while the forecasting horizon lies to the right. *(a)* When `TabPFN-TS` is not provided with frequency-related features, its predictions tend to regress toward the marginal mean of the observed history (green line). Even when frequency features are available, the model may capture dominant periodic patterns while failing to represent higher-amplitude or less frequent deviations, such as pronounced spikes. *(b)* `TabPFN-TS` exhibits difficulties in extrapolating long-term trends, particularly in short-context settings where limited historical information is available. *(c)* In some cases, the predictions of `TabPFN-TS` collapse toward the most frequent observed value in the context, leading to overly conservative forecasts that fail to reflect the recent change in dynamics. *(d)* The predicted 90% prediction intervals of `TabPFN-TS` may expand substantially, reflecting uncertainty by covering previously observed values rather than accurately capturing uncertainty in future trend evolution.

series (Hoo et al., 2025). That is, $x_{t,j} = \sin(2\pi \frac{\tau(t)}{P_j})$ or $x_{t,j} = \cos(2\pi \frac{\tau(t)}{P_j})$ where, for example, in the case of day-of-week $\tau(t) \in \{1, \cdots, 7\}$ and $P_j = 7$, and so forth. As shown in Figure 2(a), when frequency information is omitted, `TabPFN-TS` effectively reduces to estimating the mean of past observations. When the relevant frequencies are provided, `TabPFN-TS` produces accurate predictions, but it struggles to capture patterns that are not aligned with regular calendar structures.

**Weak trend extrapolation.** As already noted by Hoo et al. (2025), `TabPFN-TS` demonstrates a limited ability to extrapolate time series trends. This is seen in Figure 2(b). This phenomenon most likely results from the model's inability to consider the order of the data when estimating the trend.

**Lack of a recency bias.** `TabPFN-TS` treats all historical time points equally when making predictions. Many applications operate in environments with constant distribution shifts, e.g., the underlying data changes over time due to factors like promotions, policy changes, or macroeconomic conditions. Accurately predicting under these distribution shifts is critical for a reliable deployment of time series models. Figure 2(c) shows that `TabPFN-TS` struggles to capture a sudden uptick in demand, failing to forecast based on the most recent observations.

**Generic confidence intervals.** `TabPFN-TS` produces confidence intervals that emphasize the entire historical context rather than weighting observations according to their consistency with the prevailing trend in the

time series. Figure 2(d) clearly shows this phenomenon where the confidence interval simply reflects values obtained in the distant past. This failure undermines trust and complicates decision-making in industry critical time series applications.

## 3 ApolloPFN

ApolloPFN is a prior-data fitted network (PFN) designed for time series forecasting with exogenous covariates. Given a time series with historical observations $\mathcal{D}_{\text{train}} = (\boldsymbol{x}_t, y_t)_{t=1}^T$, where $y_t \in \mathbb{R}$ is the target and $\boldsymbol{x}_t \in \mathbb{R}^F$ are $F$-dimensional exogenous covariates observed at each time step. When future exogenous information are available, they are represented as $(\boldsymbol{x}_t)_{t=T+1}^{T+H}$ for horizon $H$. ApolloPFN produces probabilistic forecasts for a horizon $H$:

$$(y_{T+1}, \ldots, y_{T+H}) = f_\theta(y_1, \ldots, y_T, \ \boldsymbol{x}_1, \ldots, \boldsymbol{x}_{T+H}),$$

where both the context length $T$ and the covariate dimensionality $F$ may vary across instances. Crucially, ApolloPFN can anticipate covariate-driven events such as promotions, price changes, or weather shifts by conditioning on future exogenous information whereas most of the neural forecasters in the literature ignore them.

Unlike existing PFN-based forecasters that retrofit tabular models with hand-crafted time features, ApolloPFN integrates temporal inductive biases into both its training data distribution and its architecture. Below, we describe the three pillars of ApolloPFN: (i) its probabilistic inference formulation, (ii) its time-series-aware synthetic data generation, and (iii) its architectural design.

### 3.1 Prior-Data Fitted Inference for Time Series

ApolloPFN builds on the PFN paradigm introduced by Müller et al. (2022; 2025), which provides a mechanism for amortized Bayesian inference. The core idea is to train a neural network $q_\theta$ on synthetically generated datasets so that, at inference time, it directly approximates the posterior predictive distribution (PPD) without requiring explicit posterior computation or likelihood specification.

Concretely, we define a generative process that samples time series datasets $\mathcal{D} = (\boldsymbol{x}_t, y_t)_{t=1}^{T+H}$ by first sampling a latent structure $\xi \sim p(\xi)$ (in our case, a structured causal model over a directed acyclic graph) and then generating observations $(\boldsymbol{x}_t, y_t) \sim p(\boldsymbol{x}, y \mid \xi)$ with temporal dependencies. The network $q_\theta$ is trained to minimize the loss:

$$\mathcal{L}(\theta) = -\mathbb{E}_{p(\boldsymbol{x},y)} \log q_\theta(y_{\text{test}} \mid \boldsymbol{x}_{\text{test}}, \mathcal{D}_{\text{train}})$$
$$\text{i.e., } \mathcal{L}(\theta) = -\mathbb{E}_{p(\boldsymbol{x},y)} \log q_\theta\big(y_{T+1:T+H} \mid (\boldsymbol{x}_t)_{t=T+1}^{T+H}, (\boldsymbol{x}_t, y_t)_{t=1}^T\big)$$

so that $q_\theta$ directly approximates the PPD $p(y_{T+1:T+H} \mid (\boldsymbol{x}_t)_{t=T+1}^{T+H}, (\boldsymbol{x}_t, y_t)_{t=1}^T)$ (Müller et al., 2022). This circumvents the need to approximate a high-dimensional posterior $p(\xi \mid \mathcal{D}_{\text{train}})$ or to define a closed-form likelihood $p(y \mid \boldsymbol{x}, \xi)$, which is how the PPD is usually computed: $p(y_{\text{test}} \mid \boldsymbol{x}_{\text{test}}, \mathcal{D}_{\text{train}}) = \int p(y_{\text{test}} \mid \boldsymbol{x}_{\text{test}}, \xi) \, p(\xi \mid \mathcal{D}_{\text{train}}) \, d\xi$ (Murphy, 2012; Hoffman & Gelman, 2014; Wilson & Izmailov, 2020).

The key advantage of this formulation for time series is that the network learns to perform in-context inference: given a new time series at inference time, it conditions on the observed history and covariates to produce calibrated probabilistic forecasts in a single forward pass, without any gradient updates or fine-tuning.

### 3.2 Synthetic Data Generation

To train ApolloPFN, we generate synthetic time series datasets via structured causal models (SCMs) defined over directed acyclic graphs (DAGs). Each SCM specifies a generative process: given a DAG $\mathcal{G}$ with nodes $\{V_j\}$, the value at each node is determined by its parents as $V_j = f_j(V_{\text{PA}(j)}) + \epsilon_j$, where $f_j$ is a randomly sampled function (MLP with activation functions) and $\epsilon_j$ is measurement noise.

---

**Algorithm 1** Single Root Node Random Growing Network (SRNGN)

---

**Require:** $V$: total number of nodes, $\rho$ additional attachment probability
  1: Initialize graph $G$ with nodes $n = 0$, $n = 1$ and edge $(1, 0)$
  2: Initialize in-degree $k_j = 0$ for all $j \neq 0$, $k_0 = 1$
  3: **for** $n = 2, \ldots, V - 1$ **do**
  4:     Compute attachment probabilities for all nodes $i < n$
  5:     $\Pi_i = \frac{k_i + 1}{\sum_{j=0}^{n-1}(k_j + 1)}$
  6:     Select target node $t$ with probability $\Pi_t$
  7:     Select an additional source node uniformly at random from $s \in \{0, \ldots, n-1\} \setminus \{t\}$
  8:     Add new node $n$, source node connects to new node, add edge $(s, n)$
  9:     Update: $k_n \leftarrow 1$
 10:     Sample $u \sim U(0, 1)$
 11:     **if** $u < \rho$ **then**
 12:         Target connects to new node, add edge $(t, n)$
 13:         Update: $k_n \leftarrow k_n + 1$
 14:     **end if**
 15: **end for**
 16: Eliminate cycles in $G$ (if any)
 17: **return** DAG $G = (V, E)$

---

To generate a time series dataset $\mathcal{D}_{\mathcal{G}} = (\boldsymbol{x}_t, y_t)_{t=1}^N$ with $\boldsymbol{x}_t \in \mathbb{R}^F$, we first sample the total number of time steps $N \sim p(N)$ where $N = T + H$. Here, $T$ and $H$ represent the number of historical and future time steps respectively. Next, we sample the number of covariates $F \sim p(F)$. We then sample a DAG $\mathcal{G} \sim p(\mathcal{G})$ using SRNGN method (Algorithm 1) and define the SCM over it. Once we have the graph of size $|\mathcal{G}|$, we randomly select $F + 1 \leq |\mathcal{G}|$ non-root nodes and assign them as covariates $x_{t,j} = v_{t,\pi(j)}$ and target $y_t = v_{t,\pi(F+1)}$, where $\pi(\cdot)$ represents the random selection of covariates and target variable.

There are two key modifications that we make in our data generation procedure that distinguish `ApolloPFN` from prior tabular PFN approaches which are explained in detail below.

**Single Root Node Growing Network (SRNGN).** Prior work (Hollmann et al., 2025) generates DAGs using Random Growing Networks (RGN) with preferential attachment (Krapivsky & Redner, 2023) (see Algorithm 2 in Appendix A), which produce graphs with multiple root nodes and short path lengths. This is suboptimal for time series generation because many features end up disconnected (no path connects them), making them uninformative about the target.

We introduce SRNGN, a graph generation algorithm that produces graphs with a single root node and longer causal paths. SRNGN reverses the mechanics of RGN: we always incorporate new nodes by having at least one prior node connect to them, and additionally connect them to a popular node with probability $\rho$. The SRNGN construction is summarized in Algorithm 1; Figure 7 (Bottom) in Appendix A.1 illustrates the resulting graph structure. In contrast, graphs sampled via RGN (Algorithm 2 in Appendix A.1) are characterized by multiple root nodes and short path lengths (Figure 7 (Top)). Our empirical results show that SRNGN accelerates training convergence (see Figure 4).

**Time Series Root Node Excitation**. The critical distinction from tabular PFNs is how we generate observations indexed by time. Rather than sampling root node values independently for each observation (as in the i.i.d. tabular setting), we sample root node values $(v_{t,r})_{t=1}^T$ through a stochastic process that introduces temporal correlation. Specifically, we define root node values as a combination of sine and cosine functions with randomly sampled frequencies $(\phi_1^{(r)}, \phi_2^{(r)})$ and amplitudes $(\alpha_1^{(r)}, \alpha_2^{(r)})$: $v_{t,r} = \alpha_1^{(r)} \sin(\phi_1^{(r)} t) + \alpha_2^{(r)} \cos(\phi_2^{(r)} t)$, for all $t = 1, \ldots, T$. These temporally correlated root signals then propagate through the SCM graph in topological order, producing target values $y_t$ and covariates $\boldsymbol{x}_t$ that exhibit realistic temporal dependencies, including autocorrelations, quasi-periodic patterns, and covariate-driven regime changes.

We z-score the target series $(y_t)_{t=1}^T$ before passing it to the model, and invert the z-scoring when outputting the predictions $(y_t)_{t=T+1}^{T+H}$.

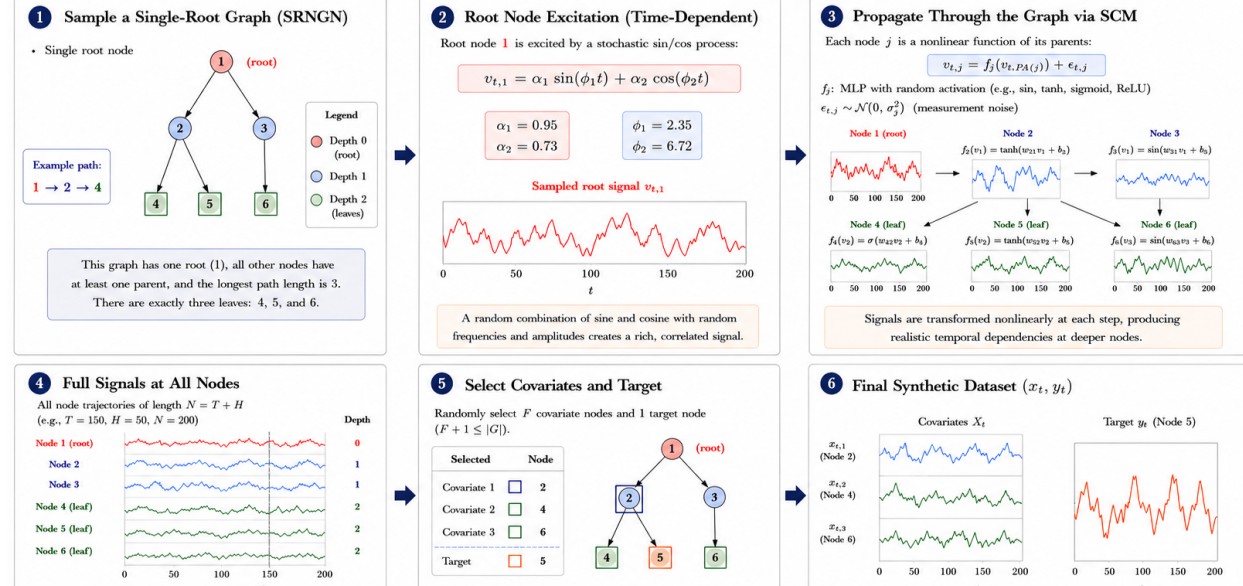

Figure 3: **Overall synthetic data generation workflow of ApolloPFN.** A stochastic root signal is propagated through a nonlinear structural causal graph to generate multivariate node trajectories, from which covariates and forecasting targets are selected to construct the final dataset $(X_t, y_t)$

### 3.3 Temporal-Feature Factorized Transformer Architecture

`ApolloPFN` employs a temporal-feature factorized transformer that processes an embedding tensor $\boldsymbol{Z} \in \mathbb{R}^{N \times F \times D}$, where $N = T + H$ is the total number of observations across both history and future horizon time steps, $F$ is the number of features (covariates plus target), and $D$ is the embedding dimension. Each layer $\ell = 1, \dots, L$ applies three operations:

$$\begin{aligned}
\boldsymbol{Z} &\leftarrow \texttt{LN}_1^{(\ell)}(\boldsymbol{Z} + \texttt{AttnFeat}^{(\ell)}(\boldsymbol{Z})) \\
\boldsymbol{Z} &\leftarrow \texttt{LN}_2^{(\ell)}(\boldsymbol{Z} + \texttt{AttnSamp}^{(\ell)}(\boldsymbol{Z})) \\
\boldsymbol{Z} &\leftarrow \texttt{LN}_3^{(\ell)}(\boldsymbol{Z} + \texttt{MLP}^{(\ell)}(\boldsymbol{Z})),
\end{aligned} \tag{1}$$

where $\texttt{LN}(\cdot)$ denotes layer normalization (Ba et al., 2016) and $\texttt{MLP}(\cdot)$ is applied along the embedding dimension. The architecture separates feature interactions from temporal interactions via two distinct attention mechanisms: $\texttt{AttnFeat}$ attends across the $F$ feature axis (learning cross-covariate relationships), while $\texttt{AttnSamp}$ attends across the temporal axis of $N$ time steps (learning temporal dependencies). This separability allows `ApolloPFN` to handle a variable number of time steps and a variable number of covariates. Appendix F provides additional details on the input embedding procedure and how it relates to Tabular PFNs such as TabPFN.

There are two key architectural modifications that distinguish `ApolloPFN` from Tabular PFNs and enable it to capture temporal structure.

**Temporal Positional Encodings.** Tabular PFNs such as `TabPFN` (Hollmann et al., 2025) use no positional encodings in `AttnSamp`, making the mechanism permutation invariant across observations. This is

appropriate for i.i.d. data but fundamentally limits the model's ability to capture temporal order. `ApolloPFN` introduces two complementary positional encoding schemes.

First, we incorporate `RoPE` embeddings (Su et al., 2023) into $\texttt{AttnSamp}^{(\ell)}(\cdot)$, which encode *relative* temporal distances. `RoPE` modifies the key-query interactions so that $\boldsymbol{q}_{t+h}^{\mathsf{T}}\boldsymbol{R}_h\boldsymbol{k}_t \to 0$ as $h \to \infty$, where $\boldsymbol{R}_h$ is a rotation matrix parameterized by the temporal lag $h$. This biases the model toward prioritizing temporally proximal observations, naturally inducing a recency effect.

Second, to incorporate an *absolute* notion of temporal position, we define sinusoidal positional encodings $\boldsymbol{\Omega} \in \mathbb{R}^{N \times D}$:

$$\Omega_{t,2d+1} = \sin\left(2\pi t \frac{2^{2d+1}}{2^{12}}\right) \quad \text{and} \quad \Omega_{t,2d} = \cos\left(2\pi t \frac{2^{2d}}{2^{12}}\right),$$

which we add to the embedding as $\boldsymbol{Z}_f \leftarrow \boldsymbol{Z}_f + \boldsymbol{\Omega}$ for all $f = 1, \ldots, F$ (see Equation 1). The combination of relative (`RoPE`) and absolute positional encodings enables `ApolloPFN` to learn both local temporal dynamics and global position-dependent patterns.

**Full Attention for Horizon-Aware Forecasting.** In tabular PFNs, test observations do not attend to each other in `AttnSamp`; each prediction is made independently as $p(y_j \mid \boldsymbol{x}_j, (\boldsymbol{x}_i, y_i)_{i=1}^N)$. This is appropriate for i.i.d. prediction but inappropriate for time series forecasting, where future exogenous covariates carry information relevant to all horizon steps.

In `ApolloPFN`, we allow all time steps (both history and horizon) to attend to each other in $\texttt{AttnSamp}^{(\ell)}(\cdot)$. This means that when forecasting, the model computes $p(y_{T+h} \mid (\boldsymbol{x}_t)_{t=T+1}^{T+H}, (\boldsymbol{x}_t, y_t)_{t=1}^T)$ for all $h = 1, \ldots, H$, where each horizon prediction is informed by the full set of future exogenous covariates. This is essential for scenarios where, e.g., a promotion scheduled at $t = T + 3$ should influence the demand forecast at $t = T + 1$ (such as anticipation or ramp up effects) or where multiple correlated events in the horizon interact.

## 4 Empirical evaluation

We comprehensively evaluate `ApolloPFN` across diverse forecasting scenarios, demonstrating strong zero-shot performance on benchmarks with exogenous information and competitive zero-shot results in the univariate setting even against models with 30 to 70× more parameters. Tables 1,2,3, and 4 summarize our main results.

**Datasets.** We evaluate `ApolloPFN` across both forecasting with exogenous variables and univariate forecasting settings. In the exogenous setting, we consider four benchmarks: (i) the M5 competition (Makridakis et al., 2022), which provides daily unit sales with price and promotional covariates aggregated to multiple temporal and geographic grains; (ii) the electricity price forecasting dataset (Lago et al., 2021b), comprising hourly prices across five European markets (NP, PJM, FR, BE, DE) with system load and generation covariates; (iii) the UCI Air Quality dataset De Vito et al. (2008), containing daily CO concentration measurements; and (iv) the Solar Energy dataset Emami (2025), with hourly generation and weather-related exogenous variables. In the univariate setting, we evaluate on the M-series suite (M1–M4) (Makridakis & Hibon, 1979; Makridakis et al., 1993; Makridakis & Hibon, 2000; Makridakis et al., 2020), which spans diverse frequencies and domains. Across all experiments, we fix the maximum context length to 512 time steps. Full dataset descriptions, including pre-processing details and exogenous feature information are provided in Appendix C.

**Compared Methods.** We compare `ApolloPFN` against methods spanning three categories: (i) statistical baselines, including AutoARIMAX and Prophet; (ii) univariate pretrained foundation models, including Sundial-Base and Chronos-Large; and (iii) pretrained foundation models with exogenous variable support, including Moirai-Large and TabPFN-TS. Notably, several of these foundation models contain 30 to 70× more parameters than `ApolloPFN`, which has only 11M parameters.[1] Detailed hyperparameter configurations of all the baselines including ApolloPFN are provided in Appendix D.

---

[1]Given the recent release of `Chronos2`, we leave a comprehensive comparison with it as future work.

### 4.1 Training Setup

**Data Generation**. The sampling procedure for our SCMs is the following. We selected the number of nodes uniformly from a minimum of 20 to a maximum of 150. Each node then contains a state of dimensionality 6 which we propagate through the graph. Moreover, when using a MLP edge we select our activation from the following options: tanh, sine, abs, identity, log, sigmoid, smooth relu, modulo and step wise (or indicator). The entries weights of the layers in the MLPs are sampled from $\mathcal{N}(0,1)$. The sample frequencies $\phi$ are sampled from $\log \phi \sim \mathcal{U}(1, 10)$ and the amplitudes $\alpha \sim \mathcal{N}(0, 1)$.

**Training**. We train our models for 400K steps using a batch size of 64 with a learning rate of 1e-4, no weight decay, 20K linear warm-up steps, and we used a cosine annealing schedule that terminates with a learning rate of 1e-6. We vary the number of samples and number of features available to the model for each batch. The number of context samples ranges from 10 to 512 and the number of features from 2 to 64 and we predict for a horizon of up to 128.

### 4.2 Zero-shot performance with exogenous variables

In this section, we evaluate and compare the zero-shot performance of the proposed `ApolloPFN` against existing methods on datasets with exogenous variables.

On the M5 aggregation benchmark (Table 1), `ApolloPFN` attains the best results at most aggregation levels, yielding an average improvement of 5.7% in RMSSE over the `TabPFN-TS`, while remaining competitive with substantially larger time-series foundation models. For electricity and domain-specific benchmarks, `ApolloPFN` achieves the strong performance on most datasets, with an average improvement of approximately 5.6% in sCRPS over `TabPFN-TS`, highlighting strong generalization across diverse domains.

| Level | Model | M5(D-B) | M5(W-B) | M5(M-B) | M5(D-S) | M5(W-S) | M5(M-S) |
|---|---|---|---|---|---|---|---|
| **State** | AutoARIMAX | 0.900 | 1.374 | 2.218 | 0.840 | 1.555 | 3.006 |
| | Prophet | 0.525 | 1.685 | 3.223 | 0.859 | 1.945 | 4.593 |
| | Sundial-Base[(0)] | 0.572 | 2.052 | 2.398 | **0.825** | 1.614 | 3.042 |
| | Chronos-Large[(0)] | 0.565 | **1.223** | 2.572 | 0.894 | 1.853 | 2.989 |
| | Moirai-Large[(†x)] | 0.794 | 1.533 | 2.336 | 0.879 | 1.681 | 2.780 |
| | TabPFN-TS[(0x)] | 0.527 | 1.251 | 2.599 | 0.883 | 1.522 | 2.864 |
| | ApolloPFN [(0x)](ours) | **0.500** | 1.275 | **2.088** | 0.875 | **1.497** | **2.721** |
| **Store** | AutoARIMAX | 0.845 | 1.775 | 2.117 | **0.811** | 1.452 | 2.463 |
| | Prophet | 1.056 | 2.070 | 3.205 | 0.822 | 1.738 | 3.507 |
| | Sundial-Base[(0)] | 1.094 | 2.124 | 2.542 | 0.815 | 1.453 | 2.422 |
| | Chronos-Large[(0)] | 0.887 | 1.704 | 2.259 | 0.880 | 1.628 | 2.476 |
| | Moirai-Large[(†x)] | 1.046 | 1.734 | 2.241 | 0.867 | 1.544 | 2.258 |
| | TabPFN-TS[(0x)] | 0.872 | 1.667 | 2.247 | 0.898 | 1.457 | 2.355 |
| | ApolloPFN [(0x)](ours) | **0.834** | **1.663** | **2.052** | 0.874 | **1.429** | **2.245** |

Table 1: **M5 Competition.** RMSSE results on M5 at state and store levels across different temporal aggregations. Lower is better. (D/W/M) denote Daily, Weekly, Monthly frequencies; (B/S) denote Category and SKU-level data. Best results are **bold**, second-best are underlined.

### 4.3 Zero-shot performance on classical univariate benchmarks

Given the limited availability of large-scale publicly accessible time series datasets, most neural forecasting models in the literature use all or a substantial portion of the M-competition data for training. Consequently, this practice complicates a fair and unbiased comparison of zero-shot model performance on these benchmarks. For completeness, however, we compare `ApolloPFN` against several of the best performing univariate

| sCRPS | DE(24) | NP(24) | FR(24) | BE(24) | PJM(24) | DE(48) | NP(48) | FR(48) | BE(48) | PJM(48) |
|---|---|---|---|---|---|---|---|---|---|---|
| AutoARIMAX | 0.156 | 0.053 | 0.120 | 0.117 | 0.200 | 0.194 | 0.065 | 0.128 | 0.149 | 0.171 |
| Prophet | 0.064 | 0.032 | 0.073 | 0.080 | 0.129 | 0.069 | 0.040 | 0.075 | 0.099 | 0.088 |
| Sundial-Base$^{(0)}$ | 0.063 | 0.022 | 0.054 | 0.053 | 0.121 | 0.081 | 0.032 | 0.053 | 0.074 | 0.057 |
| Chronos-Large$^{(0)}$ | 0.059 | 0.022 | 0.048 | 0.049 | 0.108 | 0.077 | 0.030 | **0.047** | 0.070 | 0.055 |
| Moirai-Large$^{(\dagger x)}$ | 0.102 | 0.035 | 0.081 | 0.077 | 0.140 | 0.128 | 0.049 | 0.094 | 0.105 | 0.101 |
| TabPFN-TS$^{(0x)}$ | 0.043 | **0.018** | 0.043 | 0.054 | **0.086** | **0.048** | **0.023** | 0.048 | **0.069** | **0.049** |
| ApolloPFN $^{(0x)}$(ours) | **0.042** | 0.019 | **0.040** | **0.048** | 0.091 | 0.050 | 0.026 | 0.056 | 0.070 | 0.050 |

Table 2: **Electricity Price Forecasting.** ApolloPFN achieves best or second best sCRPS results across most of the datasets and prediction horizons (24, 48). Lower values are better. The notation $^{(0x)}$ denotes zero-shot forecasters with exogenous variables; $^{(\dagger x)}$ denotes models trained with exposure to the datasets; and $^{(0)}$ denotes zero-shot univariate models without exogenous inputs. Best results are **bold**, second-best are underlined.

| sCRPS | Air Quality | Solar Weather |
|---|---|---|
| AutoARIMAX | 0.160 | 0.712 |
| Prophet | 0.121 | 0.295 |
| Sundial-Base$^{(0)}$ | 0.118 | 0.142 |
| Chronos-Large$^{(0)}$ | 0.121 | 0.157 |
| Moirai-Large$^{(\dagger x)}$ | 0.121 | 0.346 |
| TabPFN-TS$^{(0x)}$ | 0.110 | 0.135 |
| ApolloPFN $^{(0x)}$(ours) | **0.106** | **0.126** |

Table 3: **Multivariate datasets from other domains.** ApolloPFN beats other neural forecasters for series from other domains as well. sCRPS results for Air Quality dataset and Solar with Weather dataset. Lower values are better. The notation $^{(0x)}$ denotes zero-shot forecasters that leverage exogenous information; $^{(\dagger x)}$ denotes forecasters that leverage exogenous information but were exposed to the data during training; and $^{(0)}$ denotes zero-shot univariate forecasters that do not use exogenous information. Best results for each dataset are **bold**, and second best are underlined.

TSFMs. See Table 4 for a summary. Notably, ApolloPFN performs 10% better than TabPFN-TS on average across the different benchmarks.

| sCRPS | M1(M) | M1(Y) | M3(M) | M3(O) | M4(D) | M4(M) | M4(Y) | Tou(M) | Tou(Y) |
|---|---|---|---|---|---|---|---|---|---|
| Sundial-Base | 0.157 | 0.183 | 0.121 | 0.047 | 0.026 | 0.116 | 0.160 | 0.126 | 0.174 |
| Chronos-Large | 0.173 | **0.119** | 0.113 | 0.036 | 0.028 | 0.108 | **0.106** | 0.155 | **0.103** |
| Moirai-Large | **0.135** | 0.210 | **0.093** | 0.035 | 0.033 | 0.117 | 0.187 | 0.275 | 0.275 |
| TabPFN-TS | 0.169 | 0.123 | 0.106 | 0.035 | 0.027 | 0.096 | 0.115 | 0.203 | 0.146 |
| ApolloPFN (ours) | 0.152 | 0.142 | 0.094 | **0.034** | **0.023** | **0.092** | 0.113 | **0.084** | 0.137 |

Table 4: **ApolloPFN performance on the classical univariate benchmarks.** Lower values are better. Best results for each dataset are **bold**, and second best are underlined.

## 4.4 Ablation Studies

**Data Generation**. As seen in Figure 4, our use of SRNGN dramatically increases the speed at which the model starts to make accurate predictions. For Figure 4, we trained two different ApolloPFN models, one

with RGN and one with SRNGN, leaving the rest of the hyperparameters fixed. We then evaluated the performance of the model checkpoints every 10K iterations on different benchmarks. We consistently see the model trained with SRNGN achieves a better performance faster than the model trained with RGN. Additionally, we ablate the components of our data generation pipeline, including SRNGN and time-series root node excitation, to assess their individual contributions to `ApolloPFN` performance in Figure 5. We observe that the graph generation algorithm is a critical component for improving performance under a fixed training budget, as it directly shapes the complexity of relationships between covariates and targets in the data-generating prior, and consistently contributes the largest gains across multiple benchmarks.

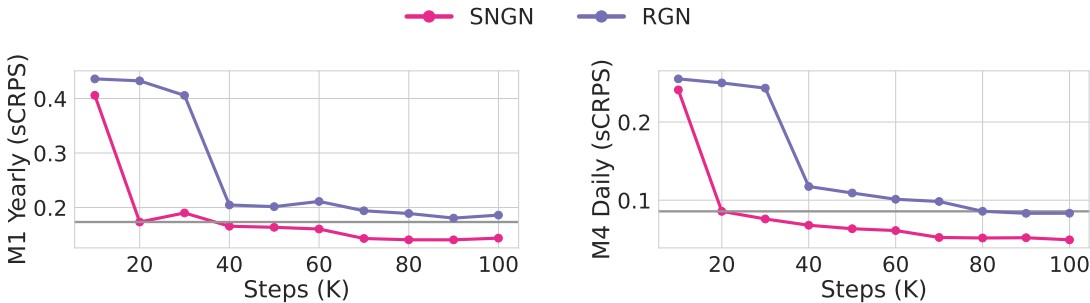

Figure 4: **SRNGN graph generation algorithm used by `ApolloPFN` accelerates learning.** We compare the test benchmark performance of our `ApolloPFN` model trained with the random growing network (RGN) algorithm and our Single Node Growing Network (SRNGN) algorithm at different training steps. With SRNGN, we achieve better performance at 20K iterations than at 80K with RGN.

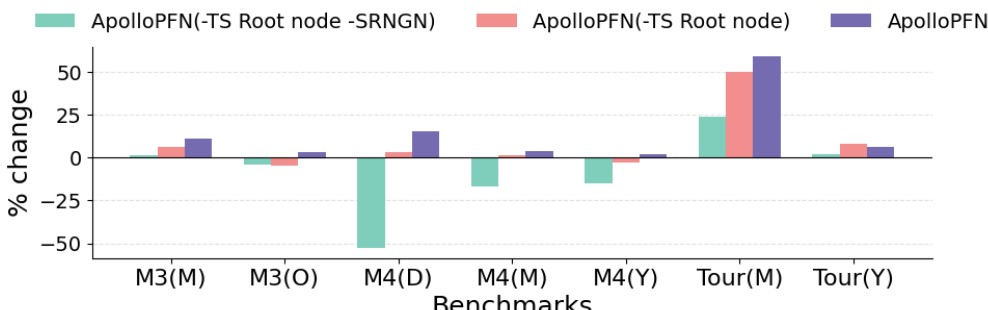

Figure 5: **Ablation of data generation components in `ApolloPFN`.** We evaluate the impact of incorporating time series root node excitation and SRNGN graph generation algorithm into the prior data generating process. Results show the relative change over the `TabPFN-TS` baseline across multiple forecasting benchmarks.

**Architecture Modifications**. To perform ablation on architectural components, we fix the control by training vanilla `TabPFN-TS` (Hollmann et al., 2025) with our time-dependent data (`ApolloPFN (-)`), then we add positional encodings (`ApolloPFN (RoPE)`), and, finally, we allow the attention mechanism to learn interactions between future exogenous samples (`ApolloPFN (RoPE+Full)`). Results appear in Figure 6, which presents the relative accuracy improvement of architectural variants with respect to `TabPFN-TS` baseline. Figure 6 shows a clear and consistent trend across test benchmarks: `ApolloPFN` achieves its strongest performance only after all proposed modifications are introduced. In particular, the most substantial improvement arises from incorporating positional encodings. `RoPE` is the primary driver of this effect, as it biases the model toward prioritizing temporally proximal observations when forming predictions. At the same time, for more complex behaviors, such as learning ordered patterns, the desired performance emerges only when all modifications are combined.

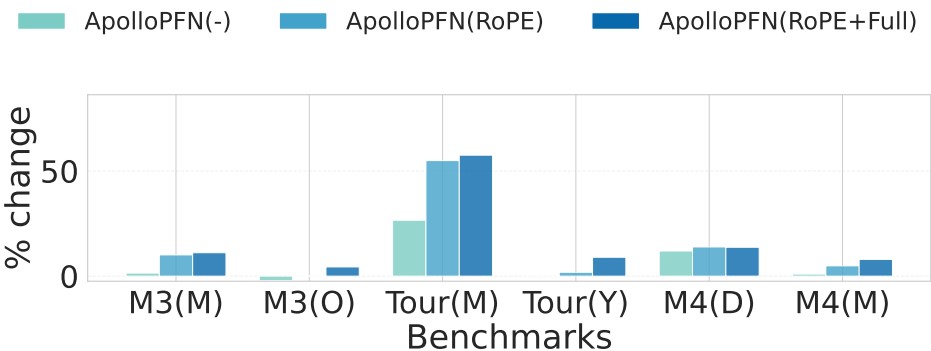

Figure 6: **Architectural modifications in `ApolloPFN` for time series forecasting.** Ablation on the use of `RoPE` positional encoding and full attention. We evaluate the effect of progressively introducing these architectural components in `ApolloPFN` and report relative change over the `TabPFN-TS` baseline across multiple forecasting benchmarks.

## 5 Conclusion

This work introduces `ApolloPFN`, a time-series specific PFN that extends the PFN paradigm to forecasting settings with temporal structure and exogenous covariates. By jointly rethinking the training distribution, architectural design, and positional bias, `ApolloPFN` moves PFNs beyond i.i.d. tabular data, toward a genuinely temporal inference model. Empirically, `ApolloPFN` improves zero-shot performance against baselines across a diverse set of forecasting benchmarks, particularly in regimes where exogenous covariates drive sharp changes, discontinuities, or heterogeneous responses across series. More broadly, our results suggest that prior-data fitted inference is a viable and powerful alternative to fine-tuning for TSFMs. By encoding temporal inductive biases directly into both the architecture and the synthetic data prior, `ApolloPFN` demonstrates how zero-shot forecasting can be made practical, scalable, and deployment-friendly. We view this work as a step toward true TSFMs that are able to "reason" over time, structure, and context in a unified way, opening the door to richer forms of probabilistic forecasting and decision-making under uncertainty.

Despite its strong empirical performance, `ApolloPFN` has several limitations that are important to acknowledge. First, the model relies on standard quadratic attention, which constrains its applicability to very long time series. This can be limiting in high-frequency settings. Second, as an in-context learning model, `ApolloPFN` may struggle in regimes with very few observations, where there is insufficient context to infer series-specific temporal structure or exogenous effects. Finally, the model's capabilities are inherently constrained by the synthetic data distribution used during training. Because the PFN paradigm learns to approximate inference under a specific prior, behaviors or dependencies that are not represented in the training data cannot be reliably recovered at test time. We expect that leveraging insights from recent work (Yu et al., 2025b;a) may help to address this issue.

These considerations naturally point to several specific directions for future work. One important avenue is the development of theoretical understanding of how the scale, diversity, and structure of the synthetic training data influence model performance and generalization. Another promising direction is the introduction of hierarchical mechanisms that allow the model to pool information across related time series, while still retaining the ability to model series-specific responses. Finally, from a systems perspective, it would be valuable to study the extent to which `ApolloPFN` can be quantized or otherwise compressed, enabling training and deployment with reduced GPU memory requirements.

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

# A Data Generation

## A.1 Vanilla Graph Generation

As explained in Section 3.1 and Section 3.2, we need to randomly generate DAGs to define diverse SCMs for our synthetic data procedure. The initial procedure to construct a graph (Hollmann et al., 2023) was through a MLP, where each node is connected to all other nodes in the next layer, and the depth of the MLP is the depth of the graph which culminates with 1 node at the end which would be the target. To illustrate, if we have a 3-layered MLP with a width of 10, then we would have a graph with $21 = 10 + 10 + 1$ nodes and $110 = 10 \times 10 + 10 \times 1$ edges (assuming that the MLP is fully connected). A step to reduce the density of the graph is to drop some edges uniformly at random or by blocks (Hollmann et al., 2023).

In Hollmann et al. (2025), the authors adopted a "more realistic" DAG generation by using a classical algorithm in the study of random networks called the random growing network with redirection (Krapivsky & Redner, 2023). This is represented in Algorithm 2.

---

**Algorithm 2** Random Growing Network (RGN), with Redirection and Preferential Attachment

---

**Require:** $V$: total number of nodes, $\rho$ redirection probability
1: Initialize graph $G$ with nodes $n = 0$, $n = 1$ and edge $(1, 0)$
2: Initialize in-degree $k_j = 0$ for all $j \neq 0$, $k_0 = 1$
3: **for** $n = 2, \ldots, V - 1$ **do**
4:     Compute attachment probabilities for all nodes $i < n$
5:     $\Pi_i = \frac{k_i + 1}{\sum_{j=0}^{n-1}(k_j + 1)}$
6:     Select target node $t$ with probability $\Pi_t$
7:     Sample $u \sim U(0, 1)$
8:     **if** $u < \rho$ **then**
9:         Connect with target, add edge $(n, t)$
10:         Update: $k_t \leftarrow k_t + 1$
11:     **else**
12:         Connect with target's only descendant, add edge $(n, d)$
13:         Update: $k_d \leftarrow k_d + 1$
14:     **end if**
15: **end for**
16: **return** DAG $G = (V, E)$

---

As illustrated in Figure 7 (Top), a characteristic of Algorithm 2 is that it generates graphs with many root nodes (as each added root node in might never get an incoming edge) and, if the redirection probability $\rho$ is high, then several of the root nodes might point to the first node. When selecting which features to use from a graph, the root nodes are always excluded (Hollmann et al., 2025), and so having a graph that has many root nodes is not necessarily optimal. Furthermore, if the graph happens to concentrate in a few nodes, then many of the features would not be related (that is, there would not be a path that connects them), making many of the features in the dataset not informative about the target.

## A.2 Proposed Graph Generation

Our proposed SRNGN is a graph generation algorithm that generates graphs with a single root node and various paths that connect new nodes. To generate our graphs to train `ApolloPFN`, we essentially reverse the mechanics of RGN with preferential attachment (Krapivsky & Redner, 2023). That is, we always incorporate nodes in a graph by having at least one prior node connect to it, and, also, we make it connect to a popular node with probability $\rho$. The SRNGN graph construction algorithm is summarized in Algorithm 1; and Figure 7 (Bottom) illustrates that this results in graphs that are connected via some path and that have only one single root node by construction. In contrast, graphs sampled via RGN (Algorithm 2) are characterized by multiple root nodes and short path lengths, as illustrated in Figure 7 (Top). Our empirical results imply

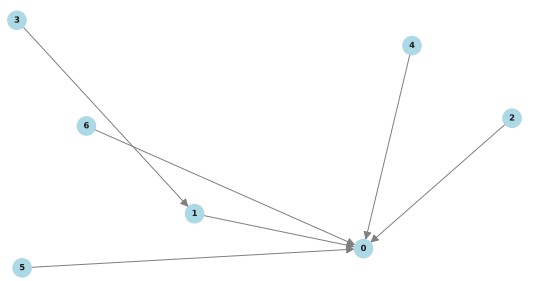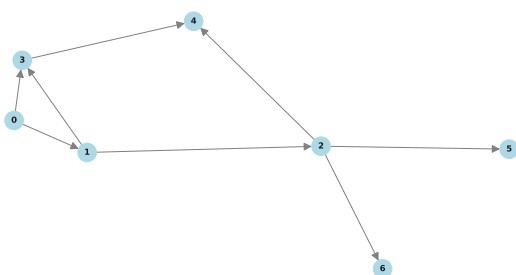

(a) Growing random network with redirection and preferential attachment (Krapivsky & Redner, 2023).

(b) Single-root growing random network inducing longer causal paths.

Figure 7: Example graphs from distinct graph generation algorithms.

that generating data using SRNGN accelerates training (see Figure 4). Similar to Hollmann et al. (2025), we sample the number of total nodes as $\log V \sim \mathcal{U}[a, b]$ and $\rho \sim \text{TrancatedGamma}(\alpha, \beta)$.

## B Metrics

We use the following set of well-established metrics to report accuracy.

**sCRPS** captures the accuracy of probabilistic predictions. This is a scale-free metric defined on quantiles $\alpha_1 < \cdots < \alpha_Q$, $\alpha_j \in (0, 1)$:

$$\text{sCRPS}(y, \hat{y}) = \frac{\sum_{t=T+1}^{T+H} \frac{2}{Q} \sum_{j=1}^{Q} \alpha_j \left(y_t - \hat{y}_t^{\alpha_j}\right)_+ + (1 - \alpha_j)\left(y_t - \hat{y}_t^{\alpha_j}\right)_-}{\sum_{t=T+1}^{T+H} |y_t|},$$

where $(\cdot)_+$ is the positive part and $(\cdot)_-$ the negative part functions. Additionally, $\hat{y}_t^{\alpha_j}$ represents the $\alpha_j$-th quantile prediction for time step $t$.

**RMSSE** is a point prediction metric used in the original M5 competition (Makridakis et al., 2022), which is defined as:

$$\text{RMSSE}(y, \hat{y}) = \frac{\frac{1}{H} \sum_{t=T+1}^{T+H} (y_t - \hat{y}_t)^2}{\frac{1}{T-1} \sum_{t=2}^{T} (y_t - y_{t-1})^2}.$$

The motivation for RMSSE is two-fold. First, it compares the predictions against a naive baseline, giving us a sense of how easy or hard it is to make predictions for a specific time series. Second, it focuses on a square error thereby penalizing models that do not capture spikes induced by exogenous variables (e.g., promotion-induced sales spikes).

## C Dataset Details

All the datasets that we used are publicly available. The M1, M3, M4, Tourism and M5 are obtained from the Monash Forecasting Repository (Godahewa et al., 2021). The Electricity Price (EPF) dataset is obtained using EPFtoolbox (Lago et al., 2021a). The UCI Air Quality and Solar Energy datasets are obtained form Fev-bench benchmark (Shchur et al., 2025). Table 5 summarizes the dataset and citations for reference.

### C.1 Electricity Price Forecasting

**Dataset Description.** The electricity price forecasting dataset (Lago et al., 2021b) consists of hourly day-ahead electricity prices for five major European markets: Nord Pool (NP), PJM (COMED zone), France (FR), Belgium (BE), and Germany (DE).

| Dataset | Source |
|---|---|
| M1 | Makridakis & Hibon (1979) |
| M3 | Makridakis & Hibon (2000) |
| M4 | Makridakis et al. (2020) |
| Tourism | Hyndman et al. (2008) |
| M5 | Makridakis et al. (2022) |
| Electricity Price | Lago et al. (2021b) |
| UCI Air Quality | De Vito et al. (2008) |
| Solar Energy | Emami (2025) |

Table 5: Data sources used for benchmarking.

**Exogenous Features.** Each market includes exogenous variables such as system load forecasts and power generation measurements, which are treated as known future covariates.

**Preprocessing.** We use the datasets in their original hourly resolution without additional preprocessing.

**Evaluation Protocol.** We evaluate each market at two forecast horizons: 24 hours and 48 hours, with a maximum context length of 512 time steps. For the 24-hour horizon, we use 50 non-overlapping evaluation windows spaced approximately 44 days apart; for the 48-hour horizon, we use 48 windows with similar spacing. The evaluation spans approximately six years per market (e.g., 2012–2017 for DE, 2011–2016 for BE and FR, 2013–2018 for NP and PJM).

## C.2 M5 Competition

**Exogenous Features.** In M5, we use price as a continuous covariate, while promotion indicators, holiday events, and SNAP calendar flags are represented as binary variables. These are treated as known future covariates.

**Preprocessing.** We use the raw daily M5 dataset and construct time series across four hierarchical levels namely {item (SKU), brand (category)} X {state, store} aggregations. In addition, we derive multiple temporal resolutions such as weekly and monthly series through temporal aggregation of the daily data. For numerical targets, we apply sum aggregation over the relevant hierarchy and time granularity. Calendar-based covariates, such as holiday indicators and SNAP dates, are defined at the daily resolution and are not aggregated across hierarchy levels. When constructing coarser temporal resolutions (e.g., weekly or monthly), these covariates are aggregated over time using a sum operation.

**Evaluation Protocol.** For each combination of temporal granularity and geographic aggregation, we use a single forecast creation date (FCD) and evaluate over the corresponding prediction horizon. Specifically, the daily variants use a 28-step horizon, the weekly variants use a 52-step horizon, and the monthly variants use a 24-step horizon.

## C.3 Domain-specific Benchmarks

**Exogenous Features.** For the UCI Air Quality dataset (De Vito et al., 2008), we follow the same procedure as in fev-bench (Shchur et al., 2025) to identify known future covariates. Specifically, we use temperature, relative humidity and absolute humidity as known future covariates and ignore the historical-only covariates present in the dataset for evaluation of all our models.

For the Solar Energy dataset (Emami, 2025), we similarly follow the fev-bench (Shchur et al., 2025) protocol to select known future covariates. We use temperature, pressure, humidity, wind speed, rain (mm over the past hour), snow (mm over the past hour), sunlight duration, day length, and sunlight ratio and use them as known future covariates and ignore the historical-only covariates present in the dataset for evaluation of all our models.

**Preprocessing.** Both datasets are obtained directly from the fev-bench benchmark in their preprocessed form. We do not apply any additional preprocessing steps.

**Evaluation Protocol.** For the UCI Air Quality dataset, we evaluate over 37 rolling windows with a 14-day forecast horizon at daily granularity. The evaluation windows are spaced 10 days apart, spanning from March 2004 to March 2005. For the Solar Energy dataset, we evaluate over 51 rolling windows with a 24-hour forecast horizon at hourly granularity. The evaluation windows are spaced approximately 40 to 46 days apart, spanning from January 2017 to August 2022. In both cases, the maximum context length is set to 512 time steps.

### C.4 M-series Benchmarks

**Dataset Description.** The M-series suite of benchmarks constitutes a comprehensive evaluation suite for the univariate case (no exogenous variables). The evaluation spans varying prediction lengths, different frequencies (hourly, daily, weekly, quarterly, yearly), and distinct data sources, resulting in widely different time series behaviors. The suite comprises four competitions: M1 (Makridakis & Hibon, 1979), M2 (Makridakis et al., 1993), M3 (Makridakis & Hibon, 2000), and M4 (Makridakis et al., 2020), which have served as consistent benchmarks for evaluating forecasting models throughout the years.

**Preprocessing and Evaluation Protocol.** We use the standard competition splits and evaluation protocols as defined in each respective competition. No additional preprocessing is applied beyond what is prescribed by the benchmark specifications. The maximum context length is set to 512 time steps across all M-series benchmarks.

## D Model Details

We compare `ApolloPFN` against methods spanning three categories:

**Statistical Baselines** *AutoARIMAX* (Hyndman et al., 2025): An automated ARIMA model with exogenous variable support that performs stepwise model selection over the space of ARIMA$(p, d, q)$ configurations. *Prophet* (Taylor & Letham, 2018): A decomposable time series model developed by Facebook that handles trends, seasonality, and holiday effects with support for additional regressors.

**Univariate Pretrained Foundation Models** *Sundial-Base* (Liu et al., 2025): A diffusion-based time series foundation model pretrained on large-scale univariate corpora. *Chronos-Large* (Ansari et al., 2024): A language-modeling-based approach to time series forecasting that tokenizes time series values and leverages transformer architectures pretrained on diverse time series data.

**Pretrained Foundation Models with Exogenous Support** *Moirai-Large* (Woo et al., 2024): A universal forecasting transformer with support for multivariate inputs and exogenous covariates, pretrained on the Large-scale Open Time Series Archive (LOTSA). *TabPFN-TS* (Hoo et al., 2025): A tabular prior-fitted network adapted for time series forecasting that natively handles exogenous features through its tabular input representation.

Model hyper parameter details for ApolloPFN and baseline models used in our evaluation pipeline are found in Table 6

---

[2]We implement the TabPFN-v2 architecture following the publicly available design specifications and initialize models directly from the released pre-trained checkpoints. We validate our implementation against the official tabpfn-time-series library, confirming agreement in outputs within numerical tolerance.

| Model | Hyperparameter Details | Code Repo |
|---|---|---|
| AutoARIMAX | Per-series fitting via `statsforecast.AutoARIMA`. Search: $p_{max}$=5, $q_{max}$=5, $d_{max}$=2, $P_{max}$=3, $Q_{max}$=3, $D_{max}$=1, season length based on frequency grain, stepwise with approximation. Covariates: exogenous regressors (NaN imputed with mean per covariate) + sin/cos seasonality features. Quantiles from prediction intervals. | `statsforecast` |
| Prophet | Per-series fitting with default hyperparameters. Interval width = 0.8. Covariates: all available features as additional regressors (NaN imputed with mean per covariate). Quantiles from MCMC posterior samples. | `prophet` |
| Sundial-base | Pre-trained `thuml/sundial-base-128m` (128M params). 100 sample trajectories via autoregressive generation. Quantiles computed empirically from samples. Univariate only (no covariates). | `HuggingFace` |
| Chronos-large | Pre-trained `amazon/chronos-t5-large` (710M params). T5 relative position encoding. Quantiles predicted directly via `predict_quantiles()`. Univariate only (no covariates). | `chronos-forecasting` |
| Moirai-large | Pre-trained `Moirai-1.1-R-large` (311M params). Patch size 16, 100 Monte Carlo samples. Quantiles computed empirically from samples. Covariates: Exogenous regressors. | `uni2ts` |
| TabPFN-TS | Pre-trained checkpoint tabpfn-v2-regressor-v2_default.ckpt from HuggingFace (11M params). 12-layer Transformer ($d$=192, 6 heads, FFN 768, no positional encoding, no dropout). Output: Bar Distribution (5000 bins). $z$-normalization per series. Covariates: Fourier features + exogenous regressors. Preprocessing: QuantileTransformer, StandardScaler, SVD. | Internal [2] |
| ApolloPFN | 12-layer Transformer ($d$=192, 6 heads, FFN 768, RoPE, full attention, no dropout, 11M params). Output: Bar Distribution (5000 bins). Training: AdamW, lr $1e^{-4}$, cosine decay ($\eta_{min}/\eta_{max}$=0.01), warmup 20k steps, 400K total steps, total batch size 64, bf16 precision. Synthetic DAG data with time series root nodes + SRNGN graph generation. $z$-normalization per series. Covariates: Exogenous regressors. Checkpoint averaging across last 10K steps as final checkpoint | Internal |

Table 6: Hyperparameter and implementation details for all evaluated models. All models use input size 512 and predict quantiles at levels $\tau \in \{0.1, 0.2, \ldots, 0.9\}$.

# E   Additional Analysis

## E.1   Robustness Analysis to Noisy and Irrelevant Exogenous Features

We evaluate the robustness of ApolloPFN under noisy and partially irrelevant exogenous inputs through controlled perturbations that vary noise type, structure, and informativeness. We consider four settings: (1) SNR-based noise injection, where Gaussian noise is added to covariates with magnitude proportional to their empirical standard deviation to control the signal-to-noise ratio; (2) partial informativeness, where covariates are replaced by a convex combination of the original signal and independent noise, governed by a mixing coefficient $\alpha$, simulating weakly predictive features; (3) irrelevant Gaussian features, where additional i.i.d. random covariates are appended; and (4) structured irrelevant features, including autoregressive (AR(1)) processes and random Fourier signals, which introduce temporally structured but task-irrelevant dependencies.

Across all settings, we follow a consistent protocol: either 50% of the original covariates are corrupted to simulate noisy or partially informative features, or 50% additional covariates are injected as irrelevant inputs.

Table 7 reports the results. ApolloPFN remains stable with minimal degradation under both unstructured (Gaussian) and structured (AR(1), Fourier) irrelevant feature injections, with only minimal deviation from the unperturbed setting. In contrast, performance degrades more noticeably under partial informativeness and low-SNR regimes, where the effective signal content of covariates is reduced. This degradation is gradual rather than abrupt, indicating that the model avoids overfitting to spurious signals even when informative structure is weakened. Overall, these experiments demonstrate that ApolloPFN is robust not only to random noise but also to structured, temporally correlated distractions that better reflect real-world exogenous covariates.

| Perturbation | Level | M5(D-B) | M5(W-B) | M5(M-B) | M5(D-S) | M5(W-S) | M5(M-S) |
|---|---|---|---|---|---|---|---|
| None | State | 0.500 | 1.275 | 2.088 | 0.875 | 1.497 | 2.721 |
| SNR (50%, medium) | State | 0.512 | 1.630 | 2.227 | 0.878 | 1.544 | 2.819 |
| SNR (50%, high) | State | 0.576 | 1.726 | 2.436 | 0.888 | 1.564 | 2.867 |
| Partial (50%) | State | 0.657 | 1.795 | 2.455 | 0.896 | 1.566 | 2.868 |
| Gaussian extra (50%) | State | 0.503 | 1.317 | 2.183 | 0.873 | 1.494 | 2.739 |
| AR(1) extra (50%) | State | 0.511 | 1.312 | 2.006 | 0.874 | 1.507 | 2.777 |
| Fourier extra (50%) | State | 0.503 | 1.368 | 2.079 | 0.873 | 1.498 | 2.724 |

Table 7: Robustness of ApolloPFN to noisy and irrelevant exogenous features under controlled perturbations across various M5-state aggregation benchmarks

## E.2   Coverage Analysis of ApolloPFN and TabPFN-TS models

We evaluate prediction interval calibration of ApolloPFN against TabPFN-TS on EPF, Solar Energy, and UCI Air Quality benchmarks. Calibration is assessed using empirical coverage at nominal levels of 20%, 40%, 60%, and 80%.

Table 8 reports the average coverage across datasets and forecasting horizons. ApolloPFN consistently achieves coverage closer to nominal levels on EPF benchmarks, particularly at mid-range intervals (40 to 60%), where calibration differences are most pronounced. In contrast, TabPFN-TS exhibits more frequent under-coverage, especially at lower and intermediate prediction intervals. ApolloPFN also demonstrates more stable behavior across datasets and horizons, with fewer large deviations from the target coverage.

To quantify calibration error, Table 9 reports the absolute deviation from nominal coverage. ApolloPFN achieves lower mean deviation across all interval levels, indicating improved and more consistent calibration. The gains are most evident in the 40% and 60% intervals, while performance at 80% remains competitive across both methods. On Solar and Air Quality datasets, both methods exhibit larger deviations at lower intervals, reflecting increased uncertainty in these settings, but ApolloPFN remains competitive overall.

| Dataset | 20% PI | | 40% PI | | 60% PI | | 80% PI | |
|---|---|---|---|---|---|---|---|---|
| | ApolloPFN | TabPFN-TS | ApolloPFN | TabPFN-TS | ApolloPFN | TabPFN-TS | ApolloPFN | TabPFN-TS |
| BE(24) | 0.214 | 0.148 | 0.434 | 0.325 | 0.622 | 0.521 | 0.813 | 0.747 |
| BE(48) | 0.229 | 0.209 | 0.438 | 0.412 | 0.620 | 0.616 | 0.793 | 0.799 |
| DE(24) | 0.212 | 0.177 | 0.422 | 0.375 | 0.608 | 0.563 | 0.813 | 0.778 |
| DE(48) | 0.206 | 0.201 | 0.398 | 0.387 | 0.608 | 0.583 | 0.816 | 0.782 |
| FR(24) | 0.242 | 0.181 | 0.480 | 0.359 | 0.672 | 0.542 | 0.833 | 0.753 |
| FR(48) | 0.224 | 0.211 | 0.427 | 0.401 | 0.614 | 0.599 | 0.785 | 0.791 |
| NP(24) | 0.181 | 0.163 | 0.389 | 0.353 | 0.578 | 0.544 | 0.773 | 0.727 |
| NP(48) | 0.237 | 0.204 | 0.428 | 0.390 | 0.620 | 0.575 | 0.830 | 0.768 |
| PJM(24) | 0.208 | 0.162 | 0.406 | 0.326 | 0.598 | 0.529 | 0.783 | 0.735 |
| PJM(48) | 0.204 | 0.177 | 0.432 | 0.349 | 0.630 | 0.560 | 0.828 | 0.777 |
| Solar | 0.462 | 0.519 | 0.650 | 0.630 | 0.779 | 0.737 | 0.898 | 0.844 |
| Air Quality | 0.224 | 0.198 | 0.418 | 0.433 | 0.648 | 0.644 | 0.785 | 0.804 |

Table 8: Average empirical coverage at different prediction interval (PI) levels for ApolloPFN and TabPFN-TS across EPF, Solar Energy, and UCI Air Quality datasets. Values closer to the nominal targets (0.2, 0.4, 0.6, 0.8) indicate better calibration.

| Dataset | 20% PI | | 40% PI | | 60% PI | | 80% PI | |
|---|---|---|---|---|---|---|---|---|
| | ApolloPFN | TabPFN-TS | ApolloPFN | TabPFN-TS | ApolloPFN | TabPFN-TS | ApolloPFN | TabPFN-TS |
| BE(24) | 0.014 | 0.053 | 0.034 | 0.075 | 0.022 | 0.079 | 0.013 | 0.053 |
| BE(48) | 0.029 | 0.009 | 0.038 | 0.012 | 0.020 | 0.016 | 0.008 | 0.001 |
| DE(24) | 0.012 | 0.023 | 0.022 | 0.025 | 0.008 | 0.037 | 0.013 | 0.022 |
| DE(48) | 0.006 | 0.001 | 0.002 | 0.013 | 0.008 | 0.017 | 0.016 | 0.018 |
| FR(24) | 0.042 | 0.019 | 0.080 | 0.041 | 0.072 | 0.058 | 0.033 | 0.048 |
| FR(48) | 0.024 | 0.011 | 0.027 | 0.001 | 0.014 | 0.001 | 0.015 | 0.009 |
| NP(24) | 0.019 | 0.038 | 0.011 | 0.048 | 0.023 | 0.056 | 0.027 | 0.073 |
| NP(48) | 0.037 | 0.004 | 0.028 | 0.010 | 0.020 | 0.025 | 0.030 | 0.032 |
| PJM(24) | 0.008 | 0.038 | 0.006 | 0.074 | 0.003 | 0.071 | 0.018 | 0.065 |
| PJM(48) | 0.004 | 0.023 | 0.032 | 0.051 | 0.030 | 0.040 | 0.028 | 0.023 |
| Solar | 0.262 | 0.319 | 0.250 | 0.230 | 0.179 | 0.137 | 0.098 | 0.044 |
| Air Quality | 0.024 | 0.002 | 0.018 | 0.033 | 0.048 | 0.044 | 0.015 | 0.004 |
| **Mean** | **0.040** | 0.045 | **0.046** | 0.051 | **0.037** | 0.048 | **0.026** | 0.033 |

Table 9: Absolute deviation from nominal coverage, computed as |empirical coverage − nominal coverage| for prediction intervals at 20%, 40%, 60%, and 80% for ApolloPFN and TabPFN-TS across EPF, Solar Energy and UCI Air Quality datasets. Lower values indicate better calibration, i.e., closer alignment between predicted and target coverage levels.

## F    TabPFN Architecture Details

Here, we provide additional details related to TabPFN architeture that might be useful for readers to understand Section 3.1.

Assume that we have the following $N_{\text{train}}$ observations for our target $(y_i)_{i=1}^{N_{\text{train}}}$, and $N = N_{\text{train}} + N_{\text{test}}$ observations for covariate information $(\boldsymbol{x}_i)_{i=1}^{N}$, where each $\boldsymbol{x}_i \in \mathbb{R}^{F'}$, and that we want to make $N_{\text{test}}$ predictions for the target $(y_i)_{i=1}^{N_{\text{test}}}$. The goal of the preprocessing step is to transform the information of $(\boldsymbol{x}_i)_{i=1}^{N}$ and $(y_i)_{i=1}^{N_{\text{train}}}$ into an embedding $\boldsymbol{Z} \in \mathbb{R}^{N \times F \times D}$, as used in Equation 1. In terms of the target, we first create a tensor $\tilde{\boldsymbol{Y}} \in \mathbb{R}^{N \times 2}$ by first z-scoring all the train targets, $\tilde{Y}_{i,1} = (y_i - \mu_{\text{train}})/\sigma_{\text{train}}$, where $\mu_{\text{train}} = \frac{1}{N_{\text{train}}} \sum_{i=1}^{N_{\text{train}}} y_i$ and $\sigma_{\text{train}}^2 = \frac{1}{N_{\text{train}}-1} \sum_{i=1}^{N_{\text{train}}} (y_i - \mu_{\text{train}})^2$, for the positions of $i = 1, \ldots, N_{\text{train}}$, and then by setting the rest of the $N_{\text{test}}$ positions $i = N_{\text{train}} + 1, \ldots, N$ as $\tilde{Y}_{i,1} = \mu_{\text{train}}$. Then, the other columns of $\tilde{\boldsymbol{Y}}$ are filled with $\tilde{Y}_{i,2} = 0$ if the entry is observed $(i = 1, \ldots, N_{\text{train}})$ and $\tilde{Y}_{i,2} = -2$ if not $(i = N_{\text{train}} + 1, \ldots, N)$. After that, we create $\boldsymbol{Y} \in \mathbb{R}^{N \times D}$ by embedding $\tilde{\boldsymbol{Y}}$ with a linear layer on a $D$ dimensional space as $\boldsymbol{Y} = \tilde{\boldsymbol{Y}} \boldsymbol{W_Y}$ where $\boldsymbol{W_Y} \in \mathbb{R}^{2 \times D}$.

An analogous procedure is done for each of the features in $\boldsymbol{x}_i \in \mathbb{R}^{F'}$ after first grouping them into pairs, as discussed in Hollmann et al. (2025). The grouping can done easily with a reshape, as follows. If we have $\tilde{\boldsymbol{X}}_i' = \boldsymbol{x}_i$, then $\tilde{\boldsymbol{X}} = \text{Reshape}(\tilde{\boldsymbol{X}}', (N, F'/2, 2))$ would have the desired effect (assuming that $F'$ is divisible by 2, else we 0-pad the feature dimension). After z-scoring each of the $f = 1, \ldots, F'/2$ features, we then compute $\boldsymbol{X} = \tilde{\boldsymbol{X}} \boldsymbol{W_X} \in \mathbb{R}^{N \times F-1 \times D}$, where $\boldsymbol{W_X} \in \mathbb{R}^{2 \times D}$ and $F = F'/2 + 1$. After the embedding $\boldsymbol{X}$ is constructed, we then add a fixed random positional encoding $\boldsymbol{\Omega} \in \mathbb{R}^{F-1 \times D}$ to each feature shared across all $N$ samples. In other words, we do $\boldsymbol{X}_i \leftarrow \boldsymbol{X}_i + \boldsymbol{\Omega}$ for all $i = 1, \ldots, N$. Finally, we set $\boldsymbol{Z} = [\boldsymbol{X}, \boldsymbol{Y}] \in \mathbb{R}^{N \times F \times D}$, which is the embedding passed to the architecture seen in Figure 8 and discussed Equation 1 in Section 3.1.

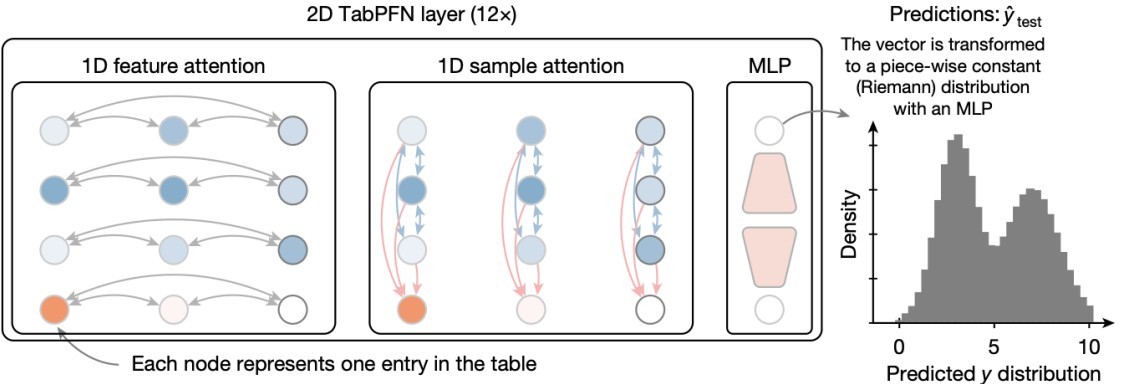

Figure 8: **How TabPFN combines attention across features and samples.** Taken from Hollmann et al. (2025), the figure illustrates the main components of the `TabPFN` architecture, discussed in Equation 1, plus the translation of the embedding into a Riemann distribution approximation of the PPD $p(y_{\text{test}}|\boldsymbol{x}_{\text{test}}, \mathcal{D}_{\text{train}})$.

The transformation of $\boldsymbol{Z} \in \mathbb{R}^{N \times F \times D}$ into the Riemman approximation of the PPD is done with another linear layer as $Z_{N_{\text{train}}:,-1,:} \boldsymbol{W_Z} \in \mathbb{R}^{N_{\text{test}} \times Q}$, where $\boldsymbol{W_Z} \in \mathbb{R}^{D \times Q}$ and $Q$ is the number of quantiles needed to compute the PPD.

