# OpenReview forum: "Time-Aware Prior Fitted Networks for Zero-Shot Forecasting with Exogenous Variables"
_TMLR — Accepted by TMLR_

### Review · Reviewer_Q6fk · 2026-03-07

**Summary Of Contributions:**

This works focuses on time-series forecasting with exogenous variables, using the Prior-Data Fitted Network (PFN). The authors argue that existing zero-shot PFN approaches do not model temporal structure and thus fail to capture e.g., time ordered patterns, recency bias, and future exogenous variables, etc. To address this, the paper proposes (1) synthetic prior data generation for constructing temporal data, and (2) introduces use of positional bias and full attention in model architecture, in order to better model the time-aware design.

**Audience:**

Yes

**Audience Explanation:**

This paper focuses on an interesting and useful topic of time-series forecasting with exogenous variables, offering potential application to real-world problems.

**Claims And Evidence:**

Yes

**Claims Explanation:**

**Strengths**

1. The paper explains the limitations of existing forecasting methods and why these limitations arise. This provides a clear motivation and builds intuition for the proposed method.
2. The method is decomposed into two main components. This makes the paper relatively easy to follow, and each part is motivated by a specific failure mode of prior work as claimed in paper. The overall design is clear with coherent narrative.
3. Experiments show empirical gains over the main PFN baseline, and is competitive with forecasting models of larger size.

**Weaknesses**

1. One main part of the method is the synthetic data generation, where temporal signals are constructed through graph-based generation (MLP, noise). It is unclear to me why this particular prior should align well with real-world problems, or real covariates such as promotions and prices.
2. The experiments present results on benchmark with exogenous variables, which may not represent real applications that often include noisy or irrelevant variables. As the method’s main contribution is learning the prior that improves exogenous forecasting, it would be valuable to test robustness / the extent to which such learned prior identifies useful covariates as opposed to simply relying on some spurious structure from the synthetic prior.
3. The proposed full attention might scale very expensively in data length. As the current experiments are set in moderate context sizes (34–512 samples, 2–64 features, and horizon of 128), it is unclear whether the performance or practicality holds for settings with longer windows or more exogenous variables.

**Requested Changes:**

I believe this would help strengthen the paper:

1. Ablations of the synthetic prior, to better validate that the proposed prior provides genuinely useful signal to learn exogenous variables, and perhaps evaluate the sensitivity of method to these choices.
2. Experiments with noisy / irrelevant exogenous features (e.g., adding controlled perturbations, or exogenous variables that are only partially informative) that better align with realistic settings
3. Since the main claim is that the method learns temporal and exogenous signals, it would be helpful to provide analysis of the learned attention patterns on benchmarks. For example, as experiments show a significant gain on benchmarks with exogenous variables, comparing attention behaviors between TabPFN-TS and the proposed method could clarify whether the introduced design causes model to use signals in the intended way. This helps to identify where the gains come from.

---

> ### Author Response · Authors · 2026-05-02
> **Response to Reviewer Q6fk [Part 1]**
>
> We thank the reviewer for these insightful suggestions, which we agree would further strengthen the analysis and interpretability of the proposed method.
>
> **(1) Ablations of the synthetic prior.**
> We agree that ablations of the synthetic prior would help further validate its role in providing useful inductive signal for learning exogenous variables, as well as assess sensitivity to design choices in the proposed data generation procedure. While this analysis would be valuable for deeper understanding, it was not included in the current experimental suite. We consider it an important direction for future work to better characterize the effect of the synthetic prior and its contribution to the overall performance.
>
> **(2) Noisy / partially irrelevant exogenous features.**
> We thank the reviewer for this suggestion and provide additional robustness experiments using controlled exogenous perturbations designed to systematically vary noise type, structure, and informativeness. Specifically, we evaluate four complementary settings: (i) **SNR-based noise injection**, where Gaussian noise is added with magnitude proportional to the empirical feature standard deviation to control signal-to-noise ratio; (ii) **partial informativeness**, where features are replaced by a convex mixture of true signal and independent noise (controlled by a mixing coefficient α), simulating weakly predictive covariates; (iii) **irrelevant Gaussian features**, where purely random features are appended to the input; and (iv) **structured irrelevant features**, including autoregressive (AR(1)) processes and random Fourier signals, which introduce temporally structured but task-irrelevant dependencies.
>
> In all settings, we follow a consistent protocol where either (i) 50% of the original covariates are corrupted to simulate noisy or partially informative features, or (ii) 50% additional covariates are injected as irrelevant features.
>
> Across these perturbations, ApolloPFN remains stable with minimal degradation under both unstructured (Gaussian) and structured (AR(1) and Fourier) noise additions, while exhibiting gradual degradation under partial informativeness and low-SNR (i.e., high noise) settings. These controlled interventions allow us to isolate different failure modes of exogenous covariates and demonstrate that the model is robust not only to random noise but also to structured spurious signals that more closely resemble real-world scenarios.
>
> | Perturbation          | Level | M5(D-B) | M5(W-B) | M5(M-B) | M5(D-S) | M5(W-S) | M5(M-S) |
> | --------------------- | ------: | ------: | ------: | ------: | ------: | ------: | ------: |
> | None                  | State |   0.500 |   1.275 |   2.088 |   0.875 |   1.497 |   2.721 |
> | snr_50_med_noise            | State |  0.512 |   1.630 |   2.227 |   0.878 |   1.544 |   2.819 |
> | snr_50_high_noise           | State |  0.576 |   1.726 |   2.436 |   0.888 |   1.564 |   2.867 |
> | partial_50            | State |  0.657 |   1.795 |   2.455 |   0.896 |   1.566 |   2.868 |
> | gaussian_extra_50     | State |  0.503 |   1.317 |   2.183 |   0.873 |   1.494 |   2.739 |
> | structured_ar1_50     | State |  0.511 |   1.312 |   2.006 |   0.874 |   1.507 |   2.777 |
> | structured_fourier_50 | State |  0.503 |   1.368 |   2.079 |   0.873 |   1.498 |   2.724 |

---

### Review · Reviewer_KQ2E · 2026-03-29

**Summary Of Contributions:**

This paper introduces ApolloPFN, a prior-data fitted network (PFN) designed for zero-shot time series forecasting that natively supports exogenous covariates. The work begins with a careful diagnosis of why existing tabular PFNs (specifically TabPFN-TS) fail on time series data: the IID assumption underlying both the synthetic training data and the permutation-invariant architecture leads to failure including inability to learn ordered patterns, weak trend extrapolation, lack of recency bias, and poorly calibrated confidence intervals. The authors introduce two ideas: (i) a new synthetic data generation procedure featuring a Single Root Node Growing Network (SRNGN) algorithm and time-dependent root node excitation via sinusoidal stochastic processes; and (ii) architectural modifications including RoPE positional encodings and full attention across future exogenous time steps.

**Key Strengths:**

1. The failure mode analysis of TabPFN-TS (Section 3, Figures 1–2) is thorough, well-illustrated, and convincingly traces each failure back to specific design choices.

2. The proposed ideas are principled and minimal: each modification is motivated by a clearly identified shortcoming, and the overall approach avoids unnecessary complexity.

3. True zero-shot evaluation is guaranteed by the synthetic-only training regime, avoiding the data contamination concerns that affect other foundation models.

**Key Weaknesses:**

1. The empirical evaluation with exogenous covariates, which is the paper's main contribution, relies on only two benchmark families (electricity pricing and M5), which is narrow.

2. Missing comparisons with classical statistical models that handle exogenous variables (ARIMAX, Prophet, gradient-boosted trees with lag features).

3. No explicit calibration analysis (reliability diagrams, coverage tables) despite criticizing TabPFN-TS's calibration.

**Audience:**

Yes

**Audience Explanation:**

The paper addresses a practically important problem: most time series foundation models cannot incorporate exogenous covariates in a zero-shot setting, yet exogenous information is essential in many applied forecasting scenarios.

**Broader Impact Concerns:**

No significant broader impact concerns.

**Claims And Evidence:**

No

**Claims Explanation:**

The methodology is sound and the approach is well-motivated, but the evidence does not fully support the breadth of the claims made. The exogenous evaluation covers only two benchmark families: electricity price forecasting and M5 retail demand. While the paper frames its contribution as a general-purpose capability, we cannot assess whether the method generalizes to other domains, covariate types, or data characteristics from this evaluation alone.

1. Electricity pricing and retail demand are both structured, high-volume domains. The paper does not evaluate on healthcare data with treatment indicators, transportation with event covariates, weather-dependent energy load forecasting, or supply chain data, among many possible alternatives.

2. The comparison set focuses on neural foundation models, omitting well-established methods that natively handle exogenous variables---ARIMAX, Prophet, and gradient-boosted tree approaches (e.g., LightGBM with lag features). In many practical settings, these methods remain strong competitors.

3. Section 3 specifically criticizes TabPFN-TS for poorly calibrated confidence intervals (Figure 2d), yet the paper does not provide reliability diagrams, coverage tables, or any calibration-specific analysis of ApolloPFN's own intervals. The sCRPS metric captures some aspects of probabilistic accuracy but does not directly measure calibration.

**Requested Changes:**

1. Include 2–3 additional benchmark families from distinct domains (e.g., healthcare, transportation, weather-dependent load forecasting) to substantiate the claim that ApolloPFN generalizes as a method for forecasting with exogenous covariates.

2. Include comparisons against at least one classical statistical method (e.g., ARIMAX or ETS with regressors) and one modern ML baseline (e.g., LightGBM with lag features, or Prophet). These are widely used in production and are natural competitors in this setting.

3. Given that Section 3 explicitly critiques TabPFN-TS's calibration, include reliability diagrams or coverage tables for ApolloPFN (and ideally baselines) to demonstrate that the claimed improvement in confidence intervals is real.

4. State the maximum context length the model can handle given quadratic attention, and clarify how much history is actually used in each benchmark. This is important for practitioners assessing applicability.

---

> ### Author Response · Authors · 2026-05-02
> **Response to Reviewer KQ2E [Part 1]**
>
> We thank the reviewer for the helpful suggestions. In response, we have made the following additions and clarifications:
>
> **(1) Additional benchmark domains.**
> We include two new datasets from distinct domains: Air Quality (UCI CO prediction) and Solar Energy with weather covariates to better evaluate generalization under heterogeneous exogenous signals. Across both datasets, ApolloPFN achieves the best performance in sCRPS, outperforming classical, neural, and foundation baselines.
>
> **sCRPS (UCI Air Quality/Solar Energy):**
> | Model                 | UCI Air Quality | Solar Energy |
> | --------------------- | ------------------: | --------------------: |
> | AutoARIMAX            |               0.160 |                 0.712 |
> | Prophet               |               0.121 |                 0.295 |
> | Sundial-Base^(0)      |               0.118 |                 0.142 |
> | Chronos-Large^(0)     |               0.121 |                 0.157 |
> | Moirai-Large^(†x)     |               0.121 |                 0.346 |
> | TabPFN-TS^(0x)        |        _0.110_ |          _0.135_ |
> | ApolloPFN^(0x) (ours) |           **0.106** |             **0.126** |
>
> **(2) Classical and modern ML baseline comparisons.**
> We add two widely used production baselines: AutoARIMAX (classical statistical forecasting with exogenous regressors) and Prophet (industry-standard ML model). Across M5 and EPF benchmarks, ApolloPFN consistently outperforms both methods across most of the dataset combinations, often by a clear margin, demonstrating strong practical competitiveness. We also expanded our existing benchmarks to evaluate multiple rolling windows with non-overlapping forecast start dates, enabling a more robust evaluation protocol.
>
> **RMSSE (M5):**
> | Level | Model            | M5(D-B)   | M5(W-B)   | M5(M-B)   | M5(D-S)   | M5(W-S)   | M5(M-S)   |
> | ----- | ---------------- | --------- | --------- | --------- | --------- | --------- | --------- |
> | State | AutoARIMAX       | 0.900     | 1.374     | *2.218*   | *0.840*   | 1.555     | 3.006     |
> | State | Prophet          | *0.525*   | 1.685     | 3.223     | 0.859     | 1.945     | 4.593     |
> | State | Sundial-Base     | 0.572     | 2.052     | 2.398     | **0.825** | 1.614     | 3.042     |
> | State | Chronos-Large    | 0.565     | **1.223** | 2.572     | 0.894     | 1.853     | 2.989     |
> | State | Moirai-Large     | 0.794     | 1.533     | 2.336     | 0.879     | 1.681     | *2.780*   |
> | State | TabPFN-TS        | 0.527     | *1.251*   | 2.599     | 0.883     | *1.522*   | 2.864     |
> | State | ApolloPFN (ours) | **0.500** | 1.275     | **2.088** | 0.875     | **1.497** | **2.721** |
>
>
> | Level | Model            | M5(D-B)   | M5(W-B)   | M5(M-B)   | M5(D-S)   | M5(W-S)   | M5(M-S)   |
> | ----- | ---------------- | --------- | --------- | --------- | --------- | --------- | --------- |
> | Store | AutoARIMAX       | *0.845*   | 1.775     | *2.117*   | **0.811** | *1.452*   | 2.463     |
> | Store | Prophet          | 1.056     | 2.070     | 3.205     | 0.822     | 1.738     | 3.507     |
> | Store | Sundial-Base     | 1.094     | 2.124     | 2.542     | *0.815*   | 1.453     | 2.422     |
> | Store | Chronos-Large    | 0.887     | 1.704     | 2.259     | 0.880     | 1.628     | 2.476     |
> | Store | Moirai-Large     | 1.046     | 1.734     | 2.241     | 0.867     | 1.544     | *2.258*   |
> | Store | TabPFN-TS        | 0.872     | *1.667*   | 2.247     | 0.898     | 1.457     | 2.355     |
> | Store | ApolloPFN (ours) | **0.834** | **1.663** | **2.052** | 0.874     | **1.429** | **2.245** |
>
>
> **sCRPS (EPF):**
> | Model                | DE(24)    | NP(24)    | FR(24)    | BE(24)    | PJM(24)   | DE(48)    | NP(48)    | FR(48)    | BE(48)    | PJM(48)   |
> | -------------------- | --------- | --------- | --------- | --------- | --------- | --------- | --------- | --------- | --------- | --------- |
> | AutoARIMAX           | 0.229     | 0.081     | 0.196     | 0.218     | 0.219     | 0.268     | 0.092     | 0.243     | 0.259     | 0.316     |
> | Prophet              | 0.064     | 0.032     | 0.073     | 0.080     | 0.129     | 0.069     | 0.040     | 0.075     | 0.099     | 0.088     |
> | Sundial-Base⁽⁰⁾      | 0.063     | 0.022     | 0.054     | 0.053     | 0.121     | 0.081     | 0.032     | 0.053     | 0.074     | 0.057     |
> | Chronos-Large⁽⁰⁾     | 0.059     | 0.022     | 0.048     | *0.049*   | 0.108     | 0.077     | 0.030     | **0.047** | *0.070*   | 0.055     |
> | Moirai-Large⁽†ˣ⁾     | 0.102     | 0.035     | 0.081     | 0.077     | 0.140     | 0.128     | 0.049     | 0.094     | 0.105     | 0.101     |
> | TabPFN-TS⁽⁰ˣ⁾        | *0.043*   | **0.018** | *0.043*   | 0.054     | **0.086** | **0.048** | **0.023** | *0.048*   | **0.069** | **0.049** |
> | ApolloPFN⁽⁰ˣ⁾ (ours) | **0.042** | *0.019*   | **0.040** | **0.048** | *0.091*   | *0.050*   | *0.026*   | 0.056     | *0.070*   | *0.050*   |

---

> ### Author Response · Authors · 2026-05-02
> **Response to Reviewer KQ2E [Part 2]**
>
> **(3) Coverage tables for ApolloPFN vs TabPFN-TS.**
> We evaluate coverage on EPF, Solar Energy, and UCI Air Quality benchmarks to support calibration improvement claims of ApolloPFN. Across 20%, 40%, 60%, and 80% prediction intervals, ApolloPFN generally achieves coverage closer to nominal levels than TabPFN-TS on EPF datasets, particularly at mid-range intervals (40–60%), where calibration differences are most evident. TabPFN-TS shows more frequent under-coverage, while ApolloPFN is more stable across datasets and horizons. Overall, the results indicate improved and more consistent calibration performance for ApolloPFN.
>
> **Average Coverage (Prediction Interval Calibration):**
> | Dataset     | 20% PI ApolloPFN | 20% PI TabPFN-TS | 40% PI ApolloPFN | 40% PI TabPFN-TS | 60% PI ApolloPFN | 60% PI TabPFN-TS | 80% PI ApolloPFN | 80% PI TabPFN-TS |
> | ----------- | ------------- | ------------- | ------------- | ------------- | ------------- | ------------- | ------------- | ------------- |
> | BE(24)      | 0.2142        | 0.1475        | 0.4342        | 0.3250        | 0.6217        | 0.5208        | 0.8125        | 0.7467        |
> | BE(48)      | 0.2292        | 0.2088        | 0.4384        | 0.4119        | 0.6198        | 0.6159        | 0.7925        | 0.7986        |
> | DE(24)      | 0.2117        | 0.1767        | 0.4217        | 0.3750        | 0.6083        | 0.5633        | 0.8125        | 0.7783        |
> | DE(48)      | 0.2057        | 0.2010        | 0.3984        | 0.3867        | 0.6076        | 0.5829        | 0.8160        | 0.7821        |
> | FR(24)      | 0.2417        | 0.1808        | 0.4800        | 0.3592        | 0.6717        | 0.5417        | 0.8325        | 0.7525        |
> | FR(48)      | 0.2244        | 0.2109        | 0.4266        | 0.4010        | 0.6141        | 0.5994        | 0.7847        | 0.7912        |
> | NP(24)      | 0.1808        | 0.1625        | 0.3892        | 0.3525        | 0.5775        | 0.5442        | 0.7733        | 0.7267        |
> | NP(48)      | 0.2365        | 0.2036        | 0.4284        | 0.3902        | 0.6202        | 0.5747        | 0.8303        | 0.7678        |
> | PJM(24)     | 0.2075        | 0.1617        | 0.4058        | 0.3258        | 0.5975        | 0.5292        | 0.7825        | 0.7350        |
> | PJM(48)     | 0.2044        | 0.1771        | 0.4319        | 0.3494        | 0.6298        | 0.5599        | 0.8281        | 0.7773        |
> | Solar       | 0.4624        | 0.5188        | 0.6495        | 0.6299        | 0.7786        | 0.7369        | 0.8979        | 0.8440        |
> | Air Quality | 0.2239        | 0.1983        | 0.4179        | 0.4328        | 0.6482        | 0.6439        | 0.7846        | 0.8038        |
>
> **Absolute Deviation from Nominal Coverage:**
> | Dataset     | 20% PI ApolloPFN | 20% PI TabPFN-TS | 40% PI ApolloPFN | 40% PI TabPFN-TS | 60% PI ApolloPFN | 60% PI TabPFN-TS | 80% PI ApolloPFN | 80% PI TabPFN-TS |
> | ----------- | ------------- | ------------- | ------------- | ------------- | ------------- | ------------- | ------------- | ------------- |
> | BE(24)      | 0.014         | 0.053         | 0.034         | 0.075         | 0.022         | 0.079         | 0.013         | 0.053         |
> | BE(48)      | 0.029         | 0.009         | 0.038         | 0.012         | 0.020         | 0.016         | 0.008         | 0.001         |
> | DE(24)      | 0.012         | 0.023         | 0.022         | 0.025         | 0.008         | 0.037         | 0.013         | 0.022         |
> | DE(48)      | 0.006         | 0.001         | 0.002         | 0.013         | 0.008         | 0.017         | 0.016         | 0.018         |
> | FR(24)      | 0.042         | 0.019         | 0.080         | 0.041         | 0.072         | 0.058         | 0.033         | 0.048         |
> | FR(48)      | 0.024         | 0.011         | 0.027         | 0.001         | 0.014         | 0.001         | 0.015         | 0.009         |
> | NP(24)      | 0.019         | 0.038         | 0.011         | 0.048         | 0.023         | 0.056         | 0.027         | 0.073         |
> | NP(48)      | 0.037         | 0.004         | 0.028         | 0.010         | 0.020         | 0.025         | 0.030         | 0.032         |
> | PJM(24)     | 0.008         | 0.038         | 0.006         | 0.074         | 0.003         | 0.071         | 0.018         | 0.065         |
> | PJM(48)     | 0.004         | 0.023         | 0.032         | 0.051         | 0.030         | 0.040         | 0.028         | 0.023         |
> | Solar       | 0.262         | 0.319         | 0.250         | 0.230         | 0.179         | 0.137         | 0.098         | 0.044         |
> | Air Quality | 0.024         | 0.002         | 0.018         | 0.033         | 0.048         | 0.044         | 0.015         | 0.004         |
> | **Mean**    | **0.040**     | 0.045     | **0.046**     | 0.051     | **0.037**     | 0.048     | **0.026**    | 0.033     |

---

> ### Author Response · Authors · 2026-05-02
> **Response to Reviewer KQ2E [Part 3]**
>
> **(4) Context length clarification.** Thank you for pointing this out. Our model uses standard quadratic attention mechanism, and in principle, the maximum context length is constrained by computational and memory considerations. In our experiments, we fix the maximum context length to 512 time steps across all selected benchmarks. We will clarify this explicitly in the paper.

---

### Review · Reviewer_nkze · 2026-04-03

**Summary Of Contributions:**

This paper proposes ApolloPFN, a time-series prior-data fitted network (PFN) for zero-shot forecasting with exogenous variables, built on the TabPFN paradigm but redesigned for temporal data. Its method has two main parts: a new synthetic data generation process that creates temporally dependent training tasks  and time-dependent root-node excitation. Overall, the paper’s key idea is to turn a tabular PFN into  time-aware PFN by changing both the training prior and the Transformer inductive bias. Although this paper proposes a new method, but it is largely an incremental improvement of exisitng TabPFN with time-series adaption.

**Additional Comments:**

The current paper looks like an incremental mofiication of existing work TabPFN which spends most of technical description on TabPFN instead of focusing on the proposed ApolloPFN.

It should reposition the proposed method with the exisitng methods it mentioned.

**Audience:**

Yes

**Audience Explanation:**

This paper proposes a novel prior-data fitted network for zero-shot time-series forecasting with exogenous variables, which are interesting for time-series domain readers.

**Claims And Evidence:**

Yes

**Claims Explanation:**

The paper’s main claims are generally supported by extensive benchmark comparisons, failure-case analyses, and ablation studies that isolate, although a small illustrative example of the synthetic graph construction would further improve clarity.

**Requested Changes:**

1. Rather than only discussing Moirai’s weakness, the discussion should be reframed to emphasize that Moirai and TabPFN/ApolloPFN represent different technical directions, i.e., large-scale pretrained forecasting versus prior-data fitted inference。

2. The paper should better position ApolloPFN as a distinct PFN design for time series, rather than mainly as an incremental adaption of TabPFN-TS. Right now, the section 2 and section mainly dicusses the TabPFN work while section 4 discusses ApolloPFN. The main technical details are revolved around TabPFN. It does not show the methodological novelty of the proposd ApolloPFN and distinguish from a straightforward engineering extension.

3. The paper introduces the synthetic graph generation process and discusses its role conceptually, but it does not provide a clear visualization or a concrete toy example showing what a sampled graph looks like. The paper could be strengthened by adding a small illustrative example and showing how time-dependent root signals are propagated through the graph to generate the synthetic time series.

4. A more complete ablation is still needed. The paper introduces two data-generation changes (SRNGN and time-dependent root-node excitation), but the current ablation directly validates only SRNGN.

In addition, the architectural study could be made more informative by separating the impact of RoPE from full attention, so that the gains can be more clearly attributed to the proposed time-aware PFN design.

---

> ### Author Response · Authors · 2026-05-02
> **Response to Reviewer nkze [Part 1]**
>
> We thank the reviewer for the constructive suggestions. We address each point below.
>
> **(1) Reframing Moirai vs. TabPFN/ApolloPFN.**
> We will clarify this point by reframing and more clearly positioning these methods as spanning two closely related but distinct paradigms: (i) large-scale pretrained forecasting models (e.g., Moirai), which rely on extensive pretraining over diverse time series corpora, and (ii) prior-data fitted inference approaches (PFNs), where prediction is performed by conditioning on the test instance through a in-context learning mechanism. At the same time, both directions can also be viewed as sharing a common goal of amortizing inference over time series tasks, differing primarily in how prior knowledge is encoded and applied.
>
> **(2) Clarifying the methodological contribution of ApolloPFN.**
> We acknowledge that the current structure places relatively more emphasis on TabPFN-TS in the early sections, which may obscure the novelty of ApolloPFN. In the revision, we will restructure Sections 2–4 to more clearly foreground ApolloPFN’s design choices.
>
>  **(3) Illustrative example for data generation.**
> We agree that the current narrative does not sufficiently illustrate the synthetic graph generation process. We note that Figure 6 in Appendix B already provides examples of sampled DAG graphs produced by our proposed graph generation algorithms; in the camera-ready version, we will complement this with a step-by-step illustration of the data generation process from root-node excitation to final time series samples.
>
> **(4) Architectural study and role of RoPE.**
> As shown in Figure 4 in Section 5.4, incorporating RoPE into the PFN design provides consistent improvements, helping quantify how much of the gain is attributable to the proposed time-aware positional modeling versus attention structure alone. We will further emphasize this interpretation in the revised discussion.

---

### Decision · Action_Editor_s7vF · 2026-05-22

**Recommendation:** Accept with minor revision

**Additional Comments:**

The initial reviews of this paper were mixed. While the reviewers appreciated the problem setting, the clear narrative motivated by specific failure cases, and the promising empirical performance, they also raised several significant concerns. These included the positioning relative to TabPFN-TS, limited insight into the new synthetic data generation procedure, missing baselines and datasets, unsupported claims, and questions regarding scalability and robustness. In their rebuttal, the authors addressed the majority of these concerns. In particular, the additional experiments were well received by the reviewers and ultimately led a majority of reviewers to recommend acceptance. I share this overall sentiment but would like to highlight some remaining issues that must be addressed before the paper can be accepted:

- The reported results do not support the state-of-the-art claims made throughout the paper. Please replace these claims with a more nuanced discussion of the empirical results.
- The description of the evaluation protocol is currently not sufficiently detailed to ensure reproducibility. Please provide all necessary information regarding data preprocessing, hyperparameter selection, covariate usage, and related implementation details (both for ApolloPFN and the baselines).
- “SRNGN” and “Time Series Root Node Excitation” are key contributions, but the manuscript does not analyze how they affect performance. Please include a performance-oriented ablation study in the same format as Figure 5.
- While some covariates (e.g., promotions) may realistically be known in advance, others (e.g., weather) may not. Please clearly specify which future covariates are conditioned on and attended to in each experiment.

Once these changes have been made, the paper will be recommended for acceptance.

**Audience:**

Yes

**Audience Explanation:**

Prior-data fitted networks (PFNs) are a well-known approach for (near) zero-shot inference with an elegant Bayesian interpretation. This paper studies PFNs for time-series forecasting with exogenous variables and shows that existing variants such as TabPFN-TS exhibit several important limitations in this setting, including reliance on frequency features, weak trend extrapolation, lack of recency bias, and poorly calibrated confidence intervals. These failure modes motivate the introduction of a new synthetic data generation procedure (→ single-root-node graphs with temporal excitation) together with architectural modifications to TabPFN-TS that enable ApolloPFN to better capture temporal structure (→ temporal positional encodings and full attention over exogenous covariates). While the proposed changes are relatively simple, they are guided by a clear narrative focused on addressing shortcomings of existing methods, and they may provide a useful foundation for future research on PFNs for temporal data.

**Claims And Evidence:**

Yes

**Claims Explanation:**

The paper compares ApolloPFN against the natural baseline TabPFN-TS across a diverse set of datasets. Although the results are not entirely conclusive, they suggest that ApolloPFN offers advantages over TabPFN-TS in several relevant settings. While evaluations against DL-based forecasting models such as DeepAR and N-BEATS are missing, the inclusion of classical approaches like Prophet and AutoARIMAX, as well as time-series foundation models such as Chronos, Moirai, and Sundial, helps position ApolloPFN within the broader state of the art. Except for classical univariate benchmarks, ApolloPFN surpasses these baselines in many relevant scenarios despite its comparatively small model size. The ablation study demonstrates the effectiveness of both relative positional encodings and full attention over exogenous covariates; however, the paper does not include a corresponding performance-oriented study of the proposed synthetic data generation procedure. The provided coverage analysis indicates more favorable calibration relative to the TabPFN-TS baseline, and the robustness experiments suggest that ApolloPFN remains stable even in the presence of noisy or irrelevant covariates.